# Presenilin/γ-secretase-dependent EphA3 processing mediates axon elongation through non-muscle myosin IIA

Míriam Javier-Torrent[1†], Sergi Marco[1†], Daniel Rocandio[2], Maria Pons-Vizcarra[1], Peter W Janes[3‡], Martin Lackmann[3§], Joaquim Egea[2], Carlos A Saura[1*]

[1]Institut de Neurociències, Department of Bioquímica i Biologia Molecular, Facultat de Medicina, Centro de Investigación Biomédica en Red Enfermedades Neurodegenerativas (CIBERNED), Universitat Autònoma de Barcelona, Barcelona, Spain; [2]Institut de Recerca Biomédica de Lleida, Universitat de Lleida, Lleida, Spain; [3]Department of Biochemistry and Molecular Biology, Monash University, Victoria, Australia

*For correspondence:
carlos.saura@uab.es

[†]These authors contributed equally to this work

Present address: [‡]Tumour Targeting Program, Olivia Newton-John Cancer Research Institute, Heidelberg, Australia

[§]Deceased

Competing interests: The authors declare that no competing interests exist.

**Abstract** EphA/ephrin signaling regulates axon growth and guidance of neurons, but whether this process occurs also independently of ephrins is unclear. We show that presenilin-1 (PS1)/γ-secretase is required for axon growth in the developing mouse brain. PS1/γ-secretase mediates axon growth by inhibiting RhoA signaling and cleaving EphA3 independently of ligand to generate an intracellular domain (ICD) fragment that reverses axon defects in PS1/γ-secretase- and EphA3-deficient hippocampal neurons. Proteomic analysis revealed that EphA3 ICD binds to non-muscle myosin IIA (NMIIA) and increases its phosphorylation (Ser1943), which promotes NMIIA filament disassembly and cytoskeleton rearrangement. PS1/γ-secretase-deficient neurons show decreased phosphorylated NMIIA and NMIIA/actin colocalization. Moreover, pharmacological NMII inhibition reverses axon retraction in PS-deficient neurons suggesting that NMIIA mediates PS/EphA3-dependent axon elongation. In conclusion, PS/γ-secretase-dependent EphA3 cleavage mediates axon growth by regulating filament assembly through RhoA signaling and NMIIA, suggesting opposite roles of EphA3 on inhibiting (ligand-dependent) and promoting (receptor processing) axon growth in developing neurons.
DOI: https://doi.org/10.7554/eLife.43646.001

## Introduction

Growth of axons towards their targets is a critical event during the establishment of neuronal connections in the developing nervous system. Growth, guidance and collapse of axons require the constant reorganization of the actin cytoskeleton at the growth cone and axons. Several signaling receptors regulate axon growth and guidance by affecting Rho family of GTPases (*Patel and Van Vactor, 2002*). Rho plays a critical role in the regulation of axon initiation, elongation, guidance and collapse by regulating the assembly, disassembly and rearrangement of the actin and microtubule cytoskeleton (*Hall and Lalli, 2010*). Rho promotes or inhibits axon elongation through its effectors mDia and Rho-associated protein kinase (ROCK), respectively. RhoA regulates negatively axon growth by activating ROCK, whereas the Rho members Rac and Cdc42 promote axon growth in hippocampal neurons (*Wang et al., 2007*). ROCK inhibits axon growth by blocking depolymerization of actin filaments indirectly by activating LIM kinase that phosphorylates ADF/cofilin, and directly by increasing phosphorylation and activating myosin II light chain (MLC) resulting in contraction of actin fibers (*Da Silva et al., 2003*; *Govek et al., 2005*). In neurons, non-muscle myosin IIA (NMIIA)/myosin-9 promotes neurite retraction caused by repulsive signals (*Wylie and Chantler, 2003*;

*Kubo et al., 2008*). Phosphorylation of NMIIA heavy chain by several kinases, including protein kinase C (PKC) or casein kinase 2 (CKII), prevents formation and/or disassembles myosin filaments (*Breckenridge et al., 2009*; *Dulyaninova and Bresnick, 2013*), although the role of NMIIA phosphorylation in axodendritic morphology is unknown.

The Eph family of receptor tyrosine kinases and their cell-attached ephrin ligands inhibit axon growth by mediating growth cone collapse through regulation of Ras and Rho GTPases (*Noren and Pasquale, 2004*). Binding of ephrins to EphA or EphB receptors of opposing cells triggers Eph phosphorylation and recruitment of cytoplasmic proteins, such as CrkII, resulting in cell-cell repulsive signals. For instance, binding of ephrinA1 to EphA4 induces Rho GEF ephexin-1 binding, which triggers RhoA activation and consequently induces growth cone collapse in hippocampal neurons (*Shamah et al., 2001*). Binding of ephrin-A5 to EphA induces growth cone collapse in a RhoA/ROCK-dependent manner (*Gao et al., 1999*; *Wahl et al., 2000*). EphA3 signaling, a relevant regulator of cell migration and neurite and axon outgrowth (*Shi et al., 2010*), regulates elongation and navigation of axons and trajectories and assembly of spinal motor neuron axons (*Marquardt et al., 2005*; *Gallarda et al., 2008*; *Nishikimi et al., 2011*).

Eph/ephrin-mediated repulsive signals in developing neurons are largely limited to cell-cell contacts during axon guidance. At present, it is unclear whether Eph receptors regulate axon elongation independently of classical ephrin signaling, but a mechanism involving shedding of ephrin ligands and receptors was recently shown to regulate EphA signaling (*Egea and Klein, 2007*). EphA activation triggers ADAM10-mediated ephrinA2 cleavage to regulate cell-cell contacts (*Hattori et al., 2000*), whereas binding of ephrinA5 to EphA3 induces an ADAM10-induced trans cleavage of the ligand (*Janes et al., 2005*). Notably, several Eph receptors and ephrin ligands are cleaved by presenilins (PS: PS1 and PS2), the catalytic component of γ-secretase involved in the pathogenesis of Alzheimer's disease (AD) (*Lleó and Saura, 2011*). EphB2 and ephrinB2 undergo sequential cleavages by metalloproteases and PS/γ-secretase, and the ephrinB2 intracellular domain (ICD) resulting from this latter cleavage regulates RhoA through Src kinase in endothelial cells (*Georgakopoulos et al., 2006*; *Litterst et al., 2007*). In addition, γ-secretase-dependent EphA4 cleavage regulates dendritic spine morphology by affecting Rac signaling (*Inoue et al., 2009*). These findings suggest that cleavage of Eph receptors regulates actin cytoskeleton through alternative mechanisms different from the classical Eph-ephrin signaling. The physiological contribution of Eph receptor shedding on axon growth and whether this event occurs independently of cell-attached ephrins remain still unclear. Here, we found that PS/γ-secretase-dependent EphA3 cleavage mediates axon elongation independently of cell-attached ephrins by promoting filament disassembly and cytoskeleton rearrangement.

## Results

### Presenilin-1/γ-secretase is essential for axon elongation in vitro and in vivo

Presenilin-1 (PS1; *Psen1*), the catalytic component of the γ-secretase complex, is mutated in most of familial Alzheimer's disease (AD) cases (*De Strooper and Annaert, 2010*). PS1 is essential for brain development as revealed by abundant brain hemorrhages, enlarged ventricles and neuron migration defects in *Psen1*[-/-] mouse embryos (*Shen et al., 1997*; *Hartmann et al., 1999*). To investigate the molecular mechanisms by which PS1/γ-secretase regulates neuron cytoskeleton in the developing brain we first performed morphological analysis of control (*Psen1*[+/+]) and *Psen1*[-/-] mouse brains at embryonic day 15.5 (E15.5) using markers of axons, including neurofilament (SMI312) and tau, and intermediate neurofilaments (nestin). Immunolabeling and confocal microscope analysis revealed reduced staining of neurofilament- and tau-stained axons and nestin filaments in the outer layer of the hippocampus and ventricular zone of *Psen1*[-/-] embryos (*Figure 1A,B*). Axon length defects in *Psen1*[-/-] brains were confirmed by retrograde DiI labeling (*Figure 1B*). Notably, neurofilament staining is largely absent in processes and restricted to the cytoplasm of doublecortin-positive immature neurons in *Psen1*[-/-] hippocampus (*Figure 1C*). The axon length defects are unlikely due to global changes on the abundance of doublecortin-positive immature neurons since number of these cells are unchanged in *Psen1*[-/-] brains (*Handler et al., 2000*). In agreement with the in vivo results, axon length was significantly reduced (~50%) in 4 days in vitro (DIV) cultured hippocampal neurons from *Psen1*[-/-] embryos, but not *Psen2*[-/-] embryos, or treated with the γ-secretase inhibitors DAPT or the

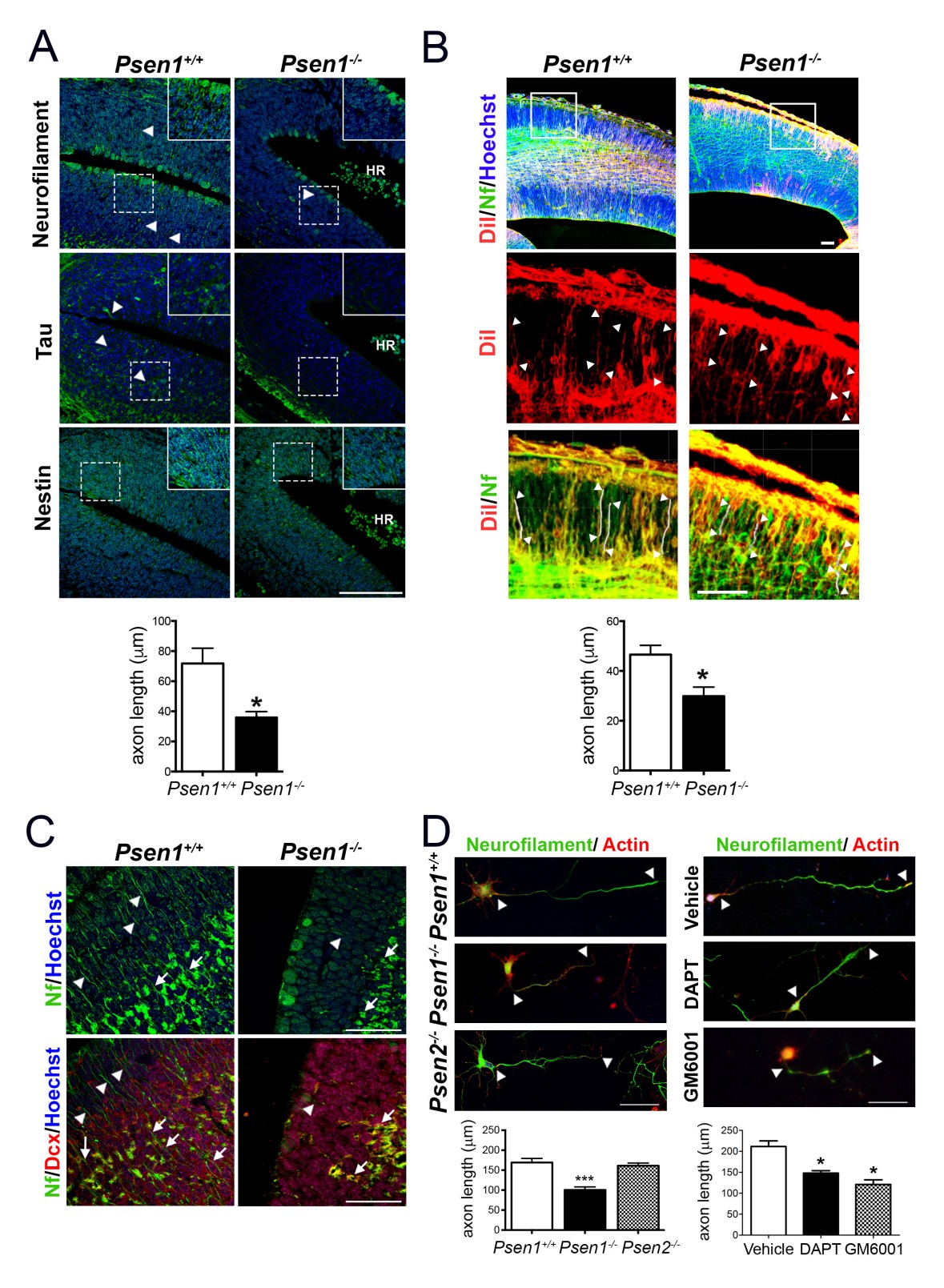

**Figure 1.** PS1/γ-secretase is required for axon growth in the developing brain. (**A**) Reduced axon length in embryonic *Psen1[-/-]* mouse brains. Confocal microscope images showing reduced axon length in the ventricular zone of *Psen1[-/-]* mouse brains (E15.5). Axons (green) are labeled with neurofilament (SM312 staining) and tau, intermediate neurofilaments with nestin, and nucleus (Hoechst) staining is shown in blue. Insets show magnified regions marked by dotted lines. *Psen1[-/-]* mouse brains show enlarged ventricles and brain hemorrhages (HR; grouped blood cells are evident). Multiple
*Figure 1 continued on next page*

*Figure 1 continued*

neurofilament-stained axons (n = 5/section; 10–15 sections/animal) were analyzed and quantified. Scale bar, 50 µm. Data are mean ± SEM (n = 3–4 animals/genotype). Unpaired two-tailed Student's *t* test: *p<0.05, compared to *Psen1*$^{+/+}$. (B) DiI labeling of axons in embryonic *Psen1*$^{+/+}$ and *Psen1*$^{-/-}$ mouse brains. Top images: confocal images showing DiI (red), neurofilament (Nf, 2H3; green) and Hoescht (blue) stainings. Insets in the top images are magnified in the middle (DiI, red) and bottom (DiI/Nf, yellow) images. Middle/bottom images: length of DiI/Nf-stained axons (in white) indicated with white arrowheads was quantified in the cortical plate of *Psen1*$^{+/+}$ and *Psen1*$^{-/-}$ embryos (E13.5; n = 3–4/group). Scale bar, 50 µm. Unpaired two-tailed Student's *t* test: *p<0.05, compared to *Psen1*$^{+/+}$. (C) Reduced axon staining in doublecortin immature neurons in the developing hippocampus of *Psen1*$^{-/-}$ embryos. *Psen1*$^{+/+}$ and *Psen1*$^{-/-}$ brain sections (E15.5) were immunostained for neurofilament (Nf, SMI321; green) and doublecortin, a marker of immature neurons (red). Confocal images show reduced axon length in doublecortin immature neurons in the inner (arrows) and outer (arrowheads) layers of *Psen1*$^{-/-}$ hippocampus. Scale bar, 50 µm. (D) Reduced axon length in primary hippocampal neurons from *Psen1*$^{-/-}$ mouse embryos. Cultured neurons (4 DIV) from control (*Psen1/2*$^{+/+}$), *Psen1*$^{-/-}$ and *Psen2*$^{-/-}$ embryos (E15.5) or neurons treated with a γ-secretase inhibitor (DAPT) or a broad spectrum metalloprotease inhibitor (GM6001) were stained with neurofilament (Nf, SMI321: green) and actin (phalloidin, red). The length of axons is indicated with white arrowheads. Multiple axons (n = 25–30/coverslip) were analyzed and quantified. Scale bars: 50 µm. Data are mean ± SEM (n = 3 experiments). One-way ANOVA followed by Bonferroni *post hoc* test: *p<0.05, compared to vehicle or control.

DOI: https://doi.org/10.7554/eLife.43646.002

The following figure supplement is available for figure 1:

**Figure supplement 1.** PS1 colocalizes and interacts with EphA3 in axons.
DOI: https://doi.org/10.7554/eLife.43646.003

broad-spectrum ADAM/matrix metalloproteinase inhibitor GM6001 (p<0.05; *Figure 1D*). These results strongly indicate that PS/γ-secretase and metalloprotease activities are required for axon growth in hippocampal neurons.

## Presenilin-1/γ-secretase-dependent EphA3 cleavage

To uncover the mechanisms responsible for PS1/γ-secretase-dependent axon elongation, we focused on EphA receptors due to its relevance in axon guidance in the developing brain (*Kania and Klein, 2016*). Quantitative real-time PCR (qRT-PCR) revealed differential expression of multiple EphA transcripts in cultured hippocampal neurons. Interestingly, *Epha3, 4, 7* and *8* mRNAs decrease significantly coinciding with last stages of axon elongation (4–7 DIV; *Figure 1—figure supplement 1A*). We focused specifically on EphA3 since: (1) EphA3 is highly expressed in axons where it regulates axon growth of hippocampal neurons in the developing brain (*Yue et al., 2002*; *Kudo et al., 2005*), (2) EphA3 protein is elevated at initial stages of axon polarization and elongation (2–4 DIV) and then it significantly decreases (*Figure 1—figure supplement 1B*), and (3) binding of ephrin-A5 to EphA3 induces the interaction of the metalloproteinase ADAM10 causing the cleavage in trans of ephrin-A5 (*Janes et al., 2005*). Notably, EphA3 is expressed as a punctuate pattern at the actin-enriched growth cones and filopodia, and along axons in hippocampal neurons, where it highly colocalizes with PS1 (~50%) (*Figure 1—figure supplement 1C*). Notably, coimmunoprecipitation assays revealed binding of PS1 to EphA3 in brain extracts of postnatal mouse brains, as well as in HEK293 cells overexpressing both proteins but not PS1 alone (*Figure 1—figure supplement 1D,E*). These results suggested binding of PS1 to EphA3 warranting investigation of EphA3 processing by PS1/γ-secretase.

To examine for a possible processing of EphA3 by PS/γ-secretase we next performed biochemical analyses using multiple anti-EphA3 antibodies in mouse brain, cultured neurons and heterologous mammalian cells. Biochemical analysis using polyclonal (C-19) and monoclonal (5E11F2) anti-C-terminal EphA3 antibodies revealed accumulation of an endogenous EphA3 C-terminal derived fragment (CTF,~49 kDa) in PS1$^{-/-}$ embryonic mouse brains and cultured neurons (*Figure 2A,B*). This suggests that this fragment could be a PS/γ-secretase substrate. DAPT increases EphA3 CTFs in EphA3-HA expressing HEK293 cells, as detected with an anti-HA antibody (*Figure 2C*). EphA3 CTFs were also present in lysates of EphA3-transfected *Psen1*$^{-/-}$/*Psen2*$^{-/-}$mouse embryonic fibroblasts (MEF) but not in control (*Psen1*$^{+/+}$/*Psen2*$^{+/+}$) or non-transfected *Psen1*$^{-/-}$/*Psen2*$^{-/-}$ MEFs (*Figure 2D*). In all cases, γ-secretase inactivation was confirmed by accumulation of N-cadherin and APP CTFs. To test for a possible metalloprotease-mediated EphA3 shedding before γ-secretase cleavage, we used the ADAM/matrix metalloproteinase inhibitors MMP9/13, GM6001 and 1,10-phenanthroline (1,10-PNT). These compounds reduced significantly accumulation of EphA3 CTFs in HEK293 cells treated with the γ-secretase inhibitors DAPT or L-685,458 indicating EphA3 shedding by metalloprotease/ADAM

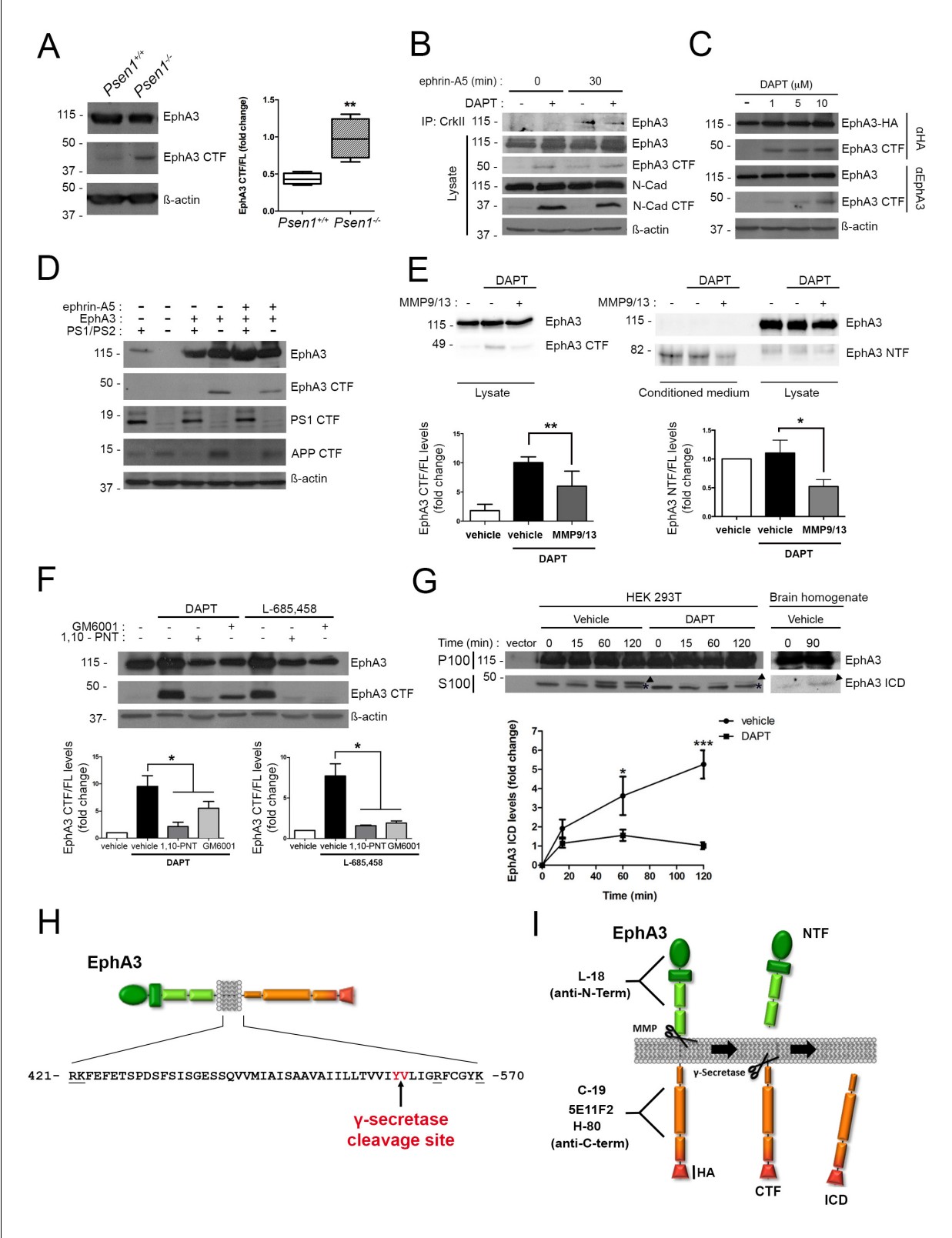

**Figure 2.** PS1/γ-secretase mediates EphA3 processing in mammalian cells. (**A**) Accumulation of EphA3 CTFs in *Psen1*[-/-] mouse brains. Western blot analysis of brain lysates from *Psen1*[+/+] and *Psen1*[-/-] mouse embryos (E15.5). Values represent mean ± SEM (n = 4); Unpaired two-tailed Student's *t* test: **p<0.01. (**B**) EphA3 CTFs accumulate in hippocampal neurons deficient in γ-secretase. Western blot analysis of EphA3 CTF (polyclonal C-19 antibody) in hippocampal neurons (4DIV) treated with γ-secretase inhibitor (DAPT) and/or ephrin-A5. (**C**) Accumulation of EphA3 CTFs in HEK293 cells treated

*Figure 2 continued on next page*

*Figure 2 continued*

with γ-secretase inhibitor. Western blot analysis of HEK293 cells expressing EphA3-HA (monoclonal anti-HA, top; 5E11F2 antibody, bottom) treated with increasing concentrations of DAPT. (D) Impaired EphA3 cleavage in PS/γ-secretase-deficient mouse embryonic fibroblasts (MEFs). Western blot analysis of EphA3 in $Psen1^{+/+}/2^{+/+}$ (PS1/PS2: +) or $Psen1^{-/-}/2^{-/-}$ (PS1/PS2: -) MEFs non-transfected (EphA3-) or transfected (EphA3+) with EphA3 in the presence or absence of clustered ephrin-A5. Reduced PS1 and accumulated APP CTFs are used as controls of decreased PS1 expression and γ-secretase-deficiency, respectively. (E) Reduced EphA3 NTFs levels in HEK293 cells treated with MMP9/13. MMP9/13 reduces significantly EphA3 CTF accumulation induced by DAPT in lysates (5E11F2 antibody; left image) and EphA3 NTFs in conditioned medium (L-18 antibody) of EphA3-overexpressing HEK293 cells (n = 5). (F) The ADAM/metalloprotease inhibitors GM6001 and 1,10-PNT abrogated EphA3 CTF accumulation induced by DAPT or L-685,458 (C-19 antibody) in HEK293 cells. (G) In vitro EphA3 ICD generation. Cell-free γ-secretase assay showing time-dependent generation of EphA3 ICD (arrowheads) in the soluble fraction (S100) of HEK293 cells transfected with EphA3 (left) and mouse brains (P2) (right). Full-length EphA3 is present only in the pellet (P100). *Indicates a degradation band that appears independently of time and treatment. In E–G), data are mean ± SD (n = 3–5 experiments). One-way ANOVA followed by Bonferroni *post hoc* test: *p<0.05, **p<0.01, ***p<0.0001, compared to vehicle or control or the indicated group. (H) Identification of the PS/γ-secretase cleavage site in EphA3. LC-MS/MS analysis of trypsin-digested gel-in samples of the EpA3 ICD band generated in vitro using the γ-secretase assay of transfected HEK293 cells identified the VLIGR peptide, demonstrating that EphA3 is cleaved at Y560 (indicated in red). Trypsin target residues are underlined. (I) Model of PS/γ-secretase-dependent EphA3 processing. EphA3 structural domains (colored boxes) and epitopes detected by antibodies used in this study. MMP, matrix metalloproteinase protein; CTF, C-terminal fragment; ICD, Intracellular Domain.

DOI: https://doi.org/10.7554/eLife.43646.004

The following figure supplement is available for figure 2:

**Figure supplement 1.** Overexpression of EphA3 mutants and *Epha3* inactivation.

DOI: https://doi.org/10.7554/eLife.43646.005

(*Figure 2E,F*). Furthermore, an EphA3 N-terminal derived fragment (NTF,~75 kDa) was significantly reduced in conditioned media in the presence of the MMP9/13 inhibitor (*Figure 2E*). An in vitro γ-secretase assay regularly used for studying APP processing was applied for detecting the EphA3 intracellular domain (ICD) (*Sastre et al., 2001*). Biochemical analysis revealed generation of EphA3 ICD (~47–49 kDa) in a time- and DAPT-dependent manner in the soluble (S100) fraction of transfected HEK293 cells, and the presence of endogenous EphA3 ICD in mouse brain lysates (*Figure 2G*).

To identify the PS/γ-secretase cleavage site in EphA3, the EphA3 ICD fragment was generated in vitro by using the γ-secretase assay and sequenced using liquid chromatography-mass spectrometry (LC-MS/MS). Proteomic analysis of trypsin-digested gel samples (~47–49 kDa band) revealed the presence of a major VLIGR peptide, corresponding to EphA3 (aa 561–565) indicating that the γ-secretase cleavage occurs at aminoacid Y560 (*Figure 2H*; *Supplementary file 1*). We detected a C-terminal peptide (NILINSNLVcK) but not a N-terminal peptide (QFAAVSITTNQAAPSPVLTIK) from that site confirming the presence of C-terminal EphA3 protein. Bioinformatic prediction of this cleavage site was also confirmed by protein sequence alignment of EPHA3 and well-established PS1/γ-secretase substrates (EPHB2, APLP1, NRXN1, CADH1, PVRL1, NOTCH1 and CD44) using ClustalW2-EMBL (http://www.ebi.ac.uk). These results demonstrate sequential cleavage of EphA3 by metalloprotease/ADAM and PS1/γ-secretase in neurons (*Figure 2I*).

## EphA3 cleavage by PS1/γ-secretase mediates axon elongation

Next, we examined the contribution of PS/γ-secretase-mediated EphA3 cleavage on regulating axon growth in hippocampal neurons. Based on the peptide sequencing analysis (*Figure 2H,I*), we generated EphA3 ICD and ΔICD mutants comprising or lacking aminoacids 561–983, respectively, and specific shRNAs to inactivate endogenous *Epha3* (*Figure 2—figure supplement 1*). EphA3 ICD-HA, ΔICD-HA and full-length were transiently expressed in vehicle or DAPT-treated $Psen1^{+/+}$ or $Psen1^{-/-}$ hippocampal neurons. Confocal imaging analysis shows that EphA3 ICD, but not EphA3 full-length or ΔICD, reversed axon length defects in $Psen1^{-/-}$ and/or DAPT-treated hippocampal neurons (*Figure 3A,B*), suggesting that PS1/γ-secretase-dependent EphA3 ICD generation is sufficient for axon growth. Moreover, EphA3 ICD recovered axon length in vehicle- or DAPT-treated EphA3-deficient neurons and reduced significantly the percentage of collapsed growth cones in $Psen1^{-/-}$ neurons (*Figure 3C*; *Figure 3—figure supplement 1*). In utero electroporation assays showed a significant decrease of axon length in the ventricular zone of $Psen1^{-/-}$ embryos (E16.5; p<0.0001; *Figure 3D*). Notably, no significant differences in axon length in EphA3 ICD expressing $Psen1^{+/+}$ and $Psen1^{-/-}$ brains were found (p=0.61). These results strongly support the relevance of EphA3 ICD

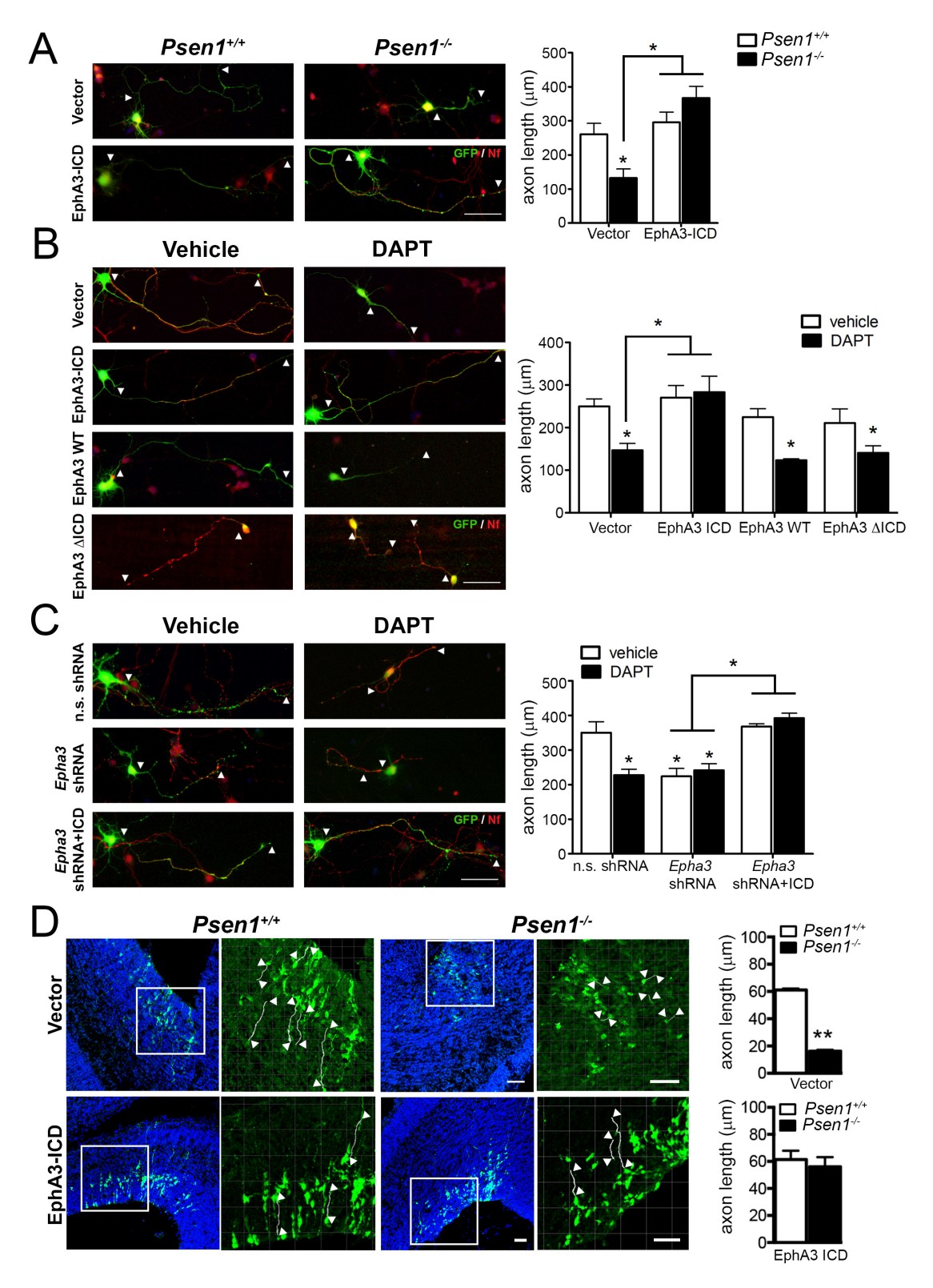

**Figure 3.** EphA3 ICD reverses axon defects in PS/γ-secretase- and EphA3-deficient neurons. (**A**) EphA3 ICD reverses axon growth defects in *Psen1*[-/-] hippocampal neurons. Immunofluorescence staining (left) and quantitative analysis (right) of axon length (indicated by arrowheads) in PS/γ-secretase-deficient neurons expressing vector or EphA3 ICD (SMI312, red; GFP, green). (**B**) EphA3 ICD reverses axon growth defects in PS/γ-secretase-deficient hippocampal neurons. EphA3 ICD (GFP, green), but not EphA3 WT or ΔICD, reverses axon growth defects in DAPT-treated hippocampal neurons.
*Figure 3 continued on next page*

*Figure 3 continued*

Axons were visualized with SMI312 (red). (C) EphA3 ICD reverses axon defects in *Epha3*-deficient neurons. EphA3 ICD (GFP, green) reverses defects in axon length (SMI312, red) both in *Epha3* ShRNA- or DAPT-treated hippocampal neurons. n.s. shRNA: non-specific scramble shRNAs. (D) In utero electroporation assays in $Psen1^{+/+}$ and $Psen1^{-/-}$ embryos. Confocal microscope images showing mCherry (red color converted to green color; top) or GFP (green; bottom) and Hoescht (blue) stainings in $Psen1^{+/+}$ and $Psen1^{-/-}$ mouse brains (E16.5) transduced with empty (mCherry; top) and EphA3 ICD (GFP; bottom) vectors. Marked square regions in the ventricular zone are magnified in the right images. Arrowheads indicate representative stained axons (white) quantified in the bar diagrams. Scale bar, 50 µm. Data in A–C) represent mean ± SEM of at least three experiments (3–6 coverslips, n = 30 neurons/coverslips). In D), data represent mean ± SEM of different sections (n = 3–9) of multiple mice. Statistics were analyzed by two-way ANOVA followed by Bonferroni *post hoc* test (A–C) or unpaired two-tailed *t* Student (D). \*\*p<0.05, \*\*p<0.0001 compared to vehicle (control), $Psen1^{+/+}$ or the indicated group.

DOI: https://doi.org/10.7554/eLife.43646.006

The following figure supplement is available for figure 3:

**Figure supplement 1.** Effect of EphA3 ICD on axon growth cone collapse.

DOI: https://doi.org/10.7554/eLife.43646.007

in mediating axon growth in the developing brain. Together, these results indicate that PS/γ-secretase regulates axon elongation through EphA3 processing.

## Ligand-independent EphA3 processing mediates axon elongation

To examine whether PS1/γ-secretase-dependent EphA3 cleavage was dependent on ligand binding, hippocampal neurons were cultured in the presence of vehicle or DAPT plus ephrin-A5, a high affinity EphA3 ligand (*Janes et al., 2005*). As previously shown (*Lawrenson et al., 2002*), ephrin-A5 efficiently enhanced EphA3 phosphorylation (Tyr779) and binding to the downstream effector CrkII (*Figures 2B* and *4A*). Notably, DAPT did not affect significantly EphA3 phosphorylation, whereas ephrin-A5 increased EphA3 signaling without major changes on EphA3 CTFs in hippocampal neurons or HEK293 (*Figures 2D* and *4A*). In agreement with previous reports (*Nishikimi et al., 2011*), ephrin-A5 reduced axon length by triggering EphA3 signaling in hippocampal neurons (*Figure 4B*). Ephrin-A5 did not cause additional effects in *Epha3* shRNA- or DAPT-treated neurons suggesting that ephrin-A5 regulates negatively axon growth through EphA3 signaling (*Figure 4B*). *Epha3* knockdown mimicked and occluded the effect of PS/γ-secretase inhibition on axon elongation in the presence or absence of ephrin-A5 (*Figure 4B*). Human EphA3 full-length, but not mutants lacking the ligand-binding (ΔLBD) or PDZ (ΔPDZ) domains, recovered axon length defects in *Epha3*-deficient hippocampal neurons (*Figure 4C,D*; and data not shown). These results strongly suggest that PS1/γ-secretase mediates axon growth through ligand-independent EphA3 signaling.

## EphA3 cleavage mediates axon elongation partially by inhibiting RhoA signaling

Since EphA signaling induces growth cone collapse in a RhoA/ROCK-dependent manner (*Wahl et al., 2000*), we next hypothesized that PS/γ-secretase-mediated EphA3 processing could regulate axon growth through RhoA signaling. To address this possibility, we examined the effects of pharmacological and genetic inactivation of RhoA signaling in axon morphology in $Psen1^{+/+}$ and $Psen1^{-/-}$ hippocampal neurons. The Rho-kinase inhibitor Y27632 induced an overall significant increase on axon growth in $Psen1^{+/+}/Psen1^{-/-}$ (p<0.0002) and Veh/DAPT-treated (p<0.0066) hippocampal neurons (*Figure 5A,B*). However, *post-hoc* analysis revealed no significant differences (p>0.05) between vehicle and Y27632 treatments in control neurons likely because RhoA signaling is inhibited during axon growth (*Ahmed et al., 2009*). Importantly, Y27632 and a dominant-negative RhoA mutant (RhoA T19N) efficiently reversed axon length defects in *Psen1*- and γ-secretase activity-deficient neurons (*Figure 5A–C*). EphA3-ICD was not able to rescue axon defects in DAPT-treated hippocampal neurons in the presence of the constitutive active RhoA mutant (RhoA Q63L), which suggests that EphA3 acts upstream of RhoA (*Figure 5D*). We next investigated the possibility that PS/γ-secretase-dependent EphA3 processing could mediate axon growth by inhibiting RhoA signaling. EphA3-ICD expression significantly reduced RhoA activity in SK-N-AS neuronal cells although it did not apparently affect RhoA activity in primary neurons in these experimental conditions (*Figure 5E,F*). Lipopolysaccharide acid (LPA) was used as a positive control because it causes growth cone collapse and neurite retraction by elevating RhoA signaling (*Hirose et al., 1998*). As

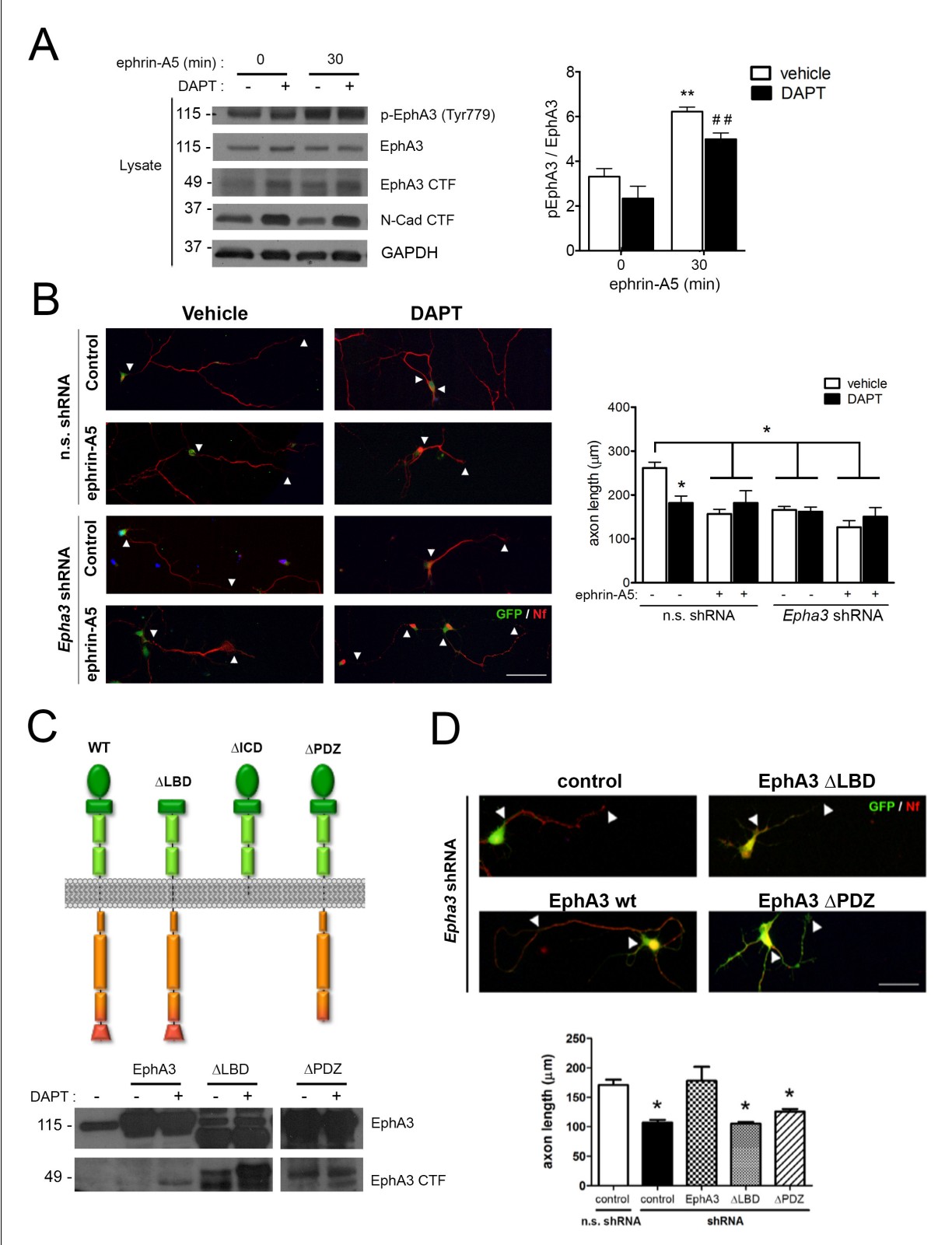

**Figure 4.** PS1/γ-secretase regulates axon growth independently of EphA3-ephrin signaling. (**A**) Inhibition of PS1/γ-secretase does not affect EphA3 phosphorylation. Biochemical analysis of total and phosphorylated (Tyr779) EphA3 and EphA3 CTFs in hippocampal cultured neurons (4 DIV) treated with vehicle (-) or the γ-secretase inhibitor DAPT (+) in the absence or presence of clustered ephrin-A5 (normalized to GAPDH). Data are mean pEphA3/EphA3 fold ± SEM (n = 3–5 experiments). Two-way ANOVA followed by Bonferroni *post hoc* test: **p<0.01, compared with control (non-stimulated)
*Figure 4 continued on next page*

*Figure 4 continued*

and, [##]p<0.01, compared to DAPT (non-stimulated). (**B**) EphA3 is required for axon elongation and ephrin-A5-mediated axon retraction. Representative immunofluorescence images (left) and quantification (right) of neurofilament-positive axons (SMI312: red) of hippocampal neurons (4 DIV) transduced with non-specific (n.s.) scramble and *Epha3* shRNAs (GFP positive, green) and treated with DAPT and/or clustered ephrin-A5. Length of axons is indicated by arrowheads. Scale bar, 50 µm. Data represent mean ± SD (n = 3 experiments; n = 30–40 cells/condition/experiment). Two-way ANOVA followed by Bonferroni *post hoc* test: *p<0.05, compared with vehicle control. (**C**) Biochemical analysis of EphA3 mutants (left) in mammalian cells. Overexpression of EphA3 WT, ΔLBD and ΔPDZ mutants in HEK293 cells. (**D**) Effect of EphA3 deleted mutants on axon growth. Immunofluorescence images (left) and quantification (right) of neurofilament-positive axons (SMI312: red; arrowheads) of hippocampal neurons (4 DIV) transduced with murine-specific *Epha3* shRNAs and the indicated EphA3 construct (GFP positive, green). Scale bar, 50 µm. Data represent mean ± SD (n = 3 experiments; n = 30–40 cells/condition/experiment). One-way ANOVA followed by Bonferroni *post hoc* test: *p<0.05, compared with vehicle control.
DOI: https://doi.org/10.7554/eLife.43646.008

previously shown in cancer cells (*Lawrenson et al., 2002*), ephrin-A5 increased RhoA activity in hippocampal neurons, whereas DAPT significantly increased and caused a synergistic effect on RhoA activation (*Figure 5F*). These results suggest that PS/γ-secretase-dependent EphA3 cleavage is unlikely to mediate axon elongation by affecting only RhoA signaling.

## EphA3 cleavage mediates axon growth via non-muscle myosin IIA

To identify EphA3 ICD binding proteins relevant for axon outgrowth, we used a proteomic approach based on anti-flag co-immunoprecipitation assays in vector- and EphA3-ICD-flag-expressing HEK293 cells. Immunoprecipitated proteins were resolved by gel electrophoresis and identified by MALDI-TOF mass spectroscopy as described (*Free et al., 2009*). Interestingly, we identified three specific prominent protein bands of ~85,~200 and~250 kDa corresponding to heat shock protein (HSP), clathrin heavy chain one and non-muscle myosin IIA heavy chain (NMIIA), respectively (*Figure 6A*; *Supplementary file 2*). We specifically focused on NMIIA/myosin 9, a protein that regulates cytoskeleton actin filament assembly and contractile forces in a variety of cell types (*Pecci et al., 2018*). In addition, casein kinase II-mediated phosphorylation (Ser1943) of NMIIA heavy chain promotes cytoskeleton rearrangement by inhibiting assembly and/or promoting disassembling of myosin/F-actin/microtubule filaments and cell motility (*Dulyaninova and Bresnick, 2013*). The physiological role of this phosphorylation in neurons is still unknown. Biochemical assays revealed that EphA3 ICD interacts with NMIIA heavy chain, and elevates axonal phosphorylated (Ser1943) NMIIA heavy chain especially in *Psen1*[-/-] hippocampal neurons (*Figure 6B,C*). Indeed, γ-secretase-deficient hippocampal neurons show decreased total phosphorylated NMIIA and phosphorylated NMIIA/actin colocalization in axons (*Figure 6D*; *Figure 6—figure supplement 1*), suggesting that γ-secretase inhibition could stabilize membrane cytoskeleton-associated NMIIA resulting in assembly or blocking disassembly of filaments. In support of this, biochemical assays show a significant decrease of soluble, increase of insoluble and unchanged total NMIIA levels in DAPT-treated hippocampal neurons (*Figure 6D*, and not shown). Indeed, a non-phosphorylated S1943A NMIIA mutant decreased significantly axon length in hippocampal neurons (*Figure 6—figure supplement 2*). We finally tested whether NMIIA function was required for axon retraction caused by PS/γ-secretase deficiency. Interestingly, blebbistatin, a specific inhibitor that binds to the myosin-ADP-Pi complex maintaining NMII in an actin-detached state (*Kovács et al., 2004*), increased axon length and occluded the effect of EphA3-ICD in DAPT-treated hippocampal neurons (*Figure 6E*). By contrast, an inactive blebbistatin enantiomer had not significant effects (*Figure 6E*). These results strongly suggest that EphA3 acts upstream of NMIIA to promote disassembly or to prevent assembly of NMIIA/actin filaments in axons.

## Discussion

EphA/ephrin signaling regulates guidance and projections of axons to proper targets by mediating cell-cell contact repulsive signals (*Kania and Klein, 2016*). Particularly, EphA3/ephrin-A5 signaling inhibits axon growth and regulates navigation of axons in developing neurons (*Gao et al., 1999*; *Marquardt et al., 2005*; *Nishikimi et al., 2011*). This is consistent with our finding that this pathway affects negatively axon growth in hippocampal neurons. However, in contrast with the established role of EphA forward signaling in growth cone collapse (*Egea and Klein, 2007*), our study provides

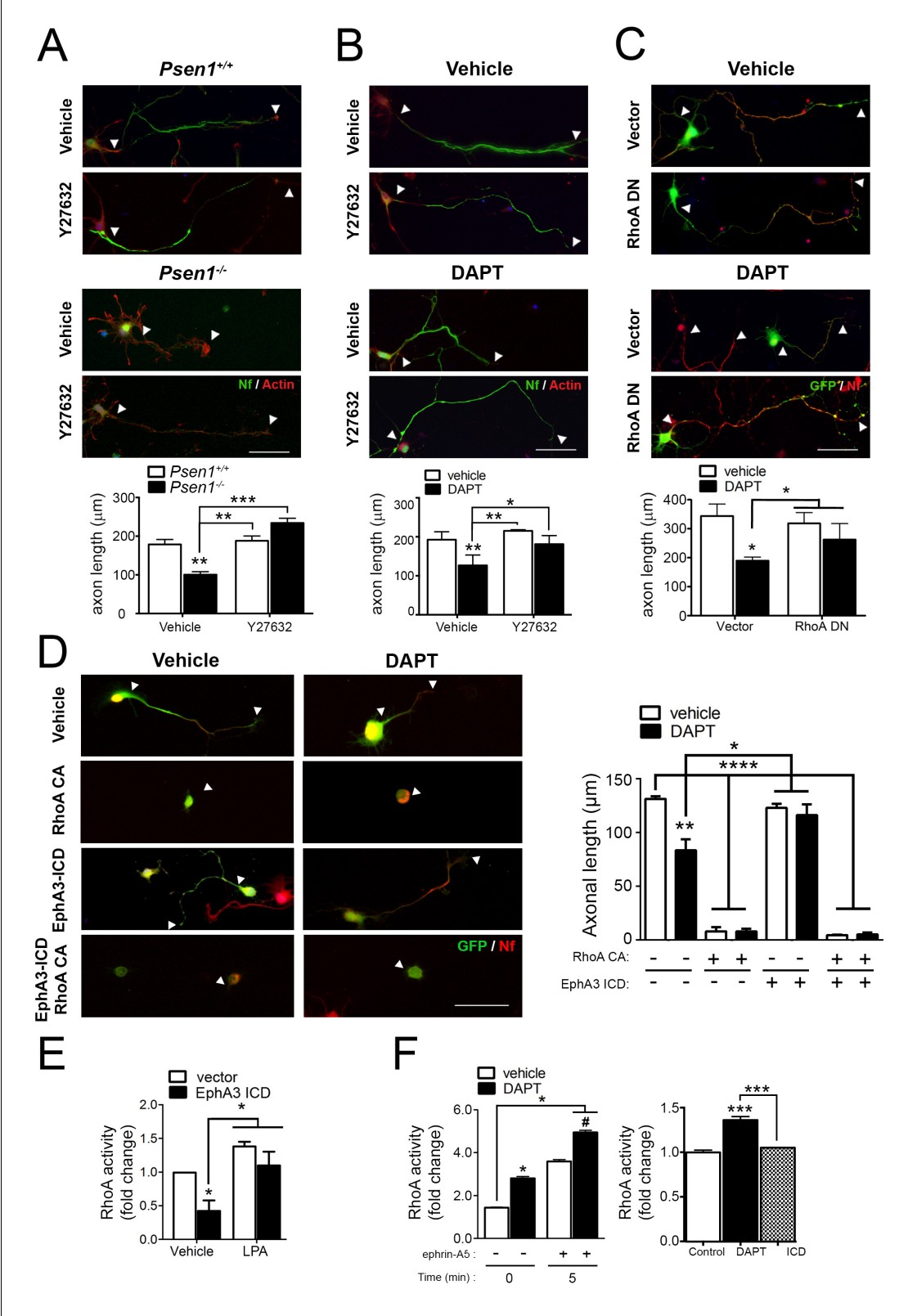

**Figure 5.** PS/γ-secretase-dependent EphA3 cleavage mediates axon elongation partially through RhoA signaling. (**A**) ROCK inhibition reverses axon length deficits in *Psen1*-/- neurons. Immunofluorescence images of hippocampal neurons from *Psen1*+/+ and *Psen1*-/- embryos treated with vehicle or the RhoA inhibitor Y27632. Neurons were stained for neurofilament (SMI312: green; arrowheads) and β-actin (red). Scale bar, 50 μm. Arrowheads indicate axon length. Data represent mean ± SEM of three independent experiments (3–6 coverslips, n = 30 neurons/coverslip). **p<0.01, ***p<0.001,
*Figure 5 continued on next page*

*Figure 5 continued*

(**B**) ROCK inhibition reverses axon length deficits in γ-secretase-deficient neurons. Images of cultured hippocampal neurons treated with vehicle or DAPT and/or the RhoA inhibitor Y27632. Neurons were stained for neurofilament (SMI312: green; arrowheads) and β-actin (red). Scale bar, 50 μm. Data represent mean ± SEM of three independent experiments (3–6 coverslips, n = 30 neurons/coverslip). *p<0.05, **p<0.01. (**C**) RhoA inhibition reverses axon length deficits γ-secretase-deficient neurons. Immunofluorescence images of hippocampal neurons stained for GFP (green) or neurofilament (SMI312, red). Neurons were transfected with GFP (vector) alone or with a RhoA dominant negative (DN) mutant (RhoA T19N) in the presence of vehicle or DAPT. Scale bar, 50 μm. Data represent mean ± SEM of at least three experiments (3–6 coverslips, n = 30 neurons/coverslip). *p<0.05. (**D**) Constitutively active RhoA affects negatively axon growth of hippocampal neurons. Neurons were transfected with GFP (vector), EphA3-ICD-GFP (EphA3 ICD) and/or a RhoA constitutive active (CA) mutant (RhoA Q63L) in the presence of vehicle or DAPT. Immunofluorescence images of hippocampal neurons stained for GFP (green) and neurofilament (SMI312, red). Scale bar, 50 μm. Data are mean ± SD of three experiments (n = 40–80 neurons/group). *p<0.05, **p<0.01, ****p<0.0001. (**E**) RhoA activity in SK-N-AS neuronal cells transfected with vector or EphA3 ICD. LPA: Lipopolysaccharide acid. Data represent mean ± SEM (n = 3–4 independent cultures). *p<0.05. (**F**) Differential effect of DAPT and EphA3 ICD in RhoA activity in cultured neurons. RhoA activity in neurons treated with vehicle or DAPT plus clustered ephrin-A5, or transduced with EphA3 ICD. Data represent mean ± SEM (n = 3). *p<0.05, ***p<0.001. Statistical analysis was performed by two-way ANOVA followed by Bonferroni *post hoc* test.
DOI: https://doi.org/10.7554/eLife.43646.009

the first evidence for a novel ligand-independent EphA3 mechanism that facilitates axon growth in neurons (*Figure 7*). First, PS1/γ-secretase-mediated EphA3 cleavage is largely independent of ligand and mediates constitutive axon growth of neurons in the developing brain. Second, *Epha3* knock-down mimicked and occluded the effect of PS/γ-secretase inhibition on axon elongation. Third, EphA3 ICD interacts with and increases phosphorylation of NMIIA heavy chain preventing assembly and/or promoting disassembly of cytoskeleton filaments in axons. These results reinforce the current view that tyrosine kinase receptor proteolytic cleavage regulates alternative intracellular signaling pathways (*Song et al., 2013*).

Our results provide evidences for a previously unappreciated role of PS in regulating axon growth. PS1 binds to and colocalizes with EphA3 in axons and growth cones of hippocampal neurons, which may explain the axonal, neuronal migration and cortical lamination defects observed in *Psen1*[-/-] embryos [(*Handler et al., 2000*); this study]. Indeed, *Psen1*[-/-] mice die after birth likely due to respiratory failures caused by incomplete lung expansion, which resembles the phenotype of *Epha3* null mice (*Shen et al., 1997*; *Vaidya et al., 2003*). Embryonic *Epha3* inactivation results in axon misrouting of callosal neurons, although no motor axon targeting abnormalities are observed in *Epha3*[-/-] mice (*Vaidya et al., 2003*; *Nishikimi et al., 2011*). By contrast, *Epha3/Epha4* deletion causes disturbances in motor axon guidance and sensory-motor neuron assemblies (*Gallarda et al., 2008*). Similarly, it is well established that PS1 regulates axon guidance and kinesin-mediated axonal transport of motor neurons (*Kamal et al., 2001*; *Bai et al., 2011*). Loss of function mutations in the presenilin genes *sel-12* and *hop-1* result in abnormal axonal projections in *C. elegans*, an effect attributed to altered Notch signaling (*Wittenburg et al., 2000*). While our study indicates that PS1 promotes axonal growth, a recent study demonstrates that loss of PS1 function results in neurite outgrowth by increasing APP intracellular fragments and activating CREB signaling (*Deyts et al., 2016*). Interestingly, inactivation of β-secretase (BACE1), which is involved in APP processing, also causes axon guidance defects mediated by altered processing of the neural cell adhesion molecule close homolog of L1 (CHL1) (*Hitt et al., 2012*). Moreover, proteolytic processing of neuregulin-1 by BACE and ADAM17 proteases mediates myelination of axons in the peripheral nervous system (*Hu et al., 2006*; *Willem et al., 2006*). Of interest, familial-AD linked PS1 mutations, acting through a partial loss of function mechanism (*De Strooper, 2007*; *Shen and Kelleher, 2007*), cause aberrant APP-dependent axodendritic growth in cultured neurons (*Pigino et al., 2001*; *Deyts et al., 2016*). Despite these evidences, the PS1/γ-secretase-dependent mechanisms mediating cytoskeleton rearrangement during axon growth have been elusive.

Our results reveal that PS1/γ-secretase-dependent EphA3 cleavage mediates axon elongation. PS1/γ-secretase-regulated EphA3 processing is essential for constitutive axon elongation as suggested by the mimicking and occluding effects of *Epha3* inactivation on axon elongation in PS-deficient neurons. Indeed, EphA3 ICD expression recovered axon defects of PS1-deficient neurons in vitro (cultured neurons) and in vivo (embryonic brain). PS1/γ-secretase/EphA3-dependent axon growth contrasts with the classical role of ligand-induced EphA3 signaling in axon retraction. In agreement, both ligand-induced signaling and blocking ligand-independent EphA3 cleavage (e.g

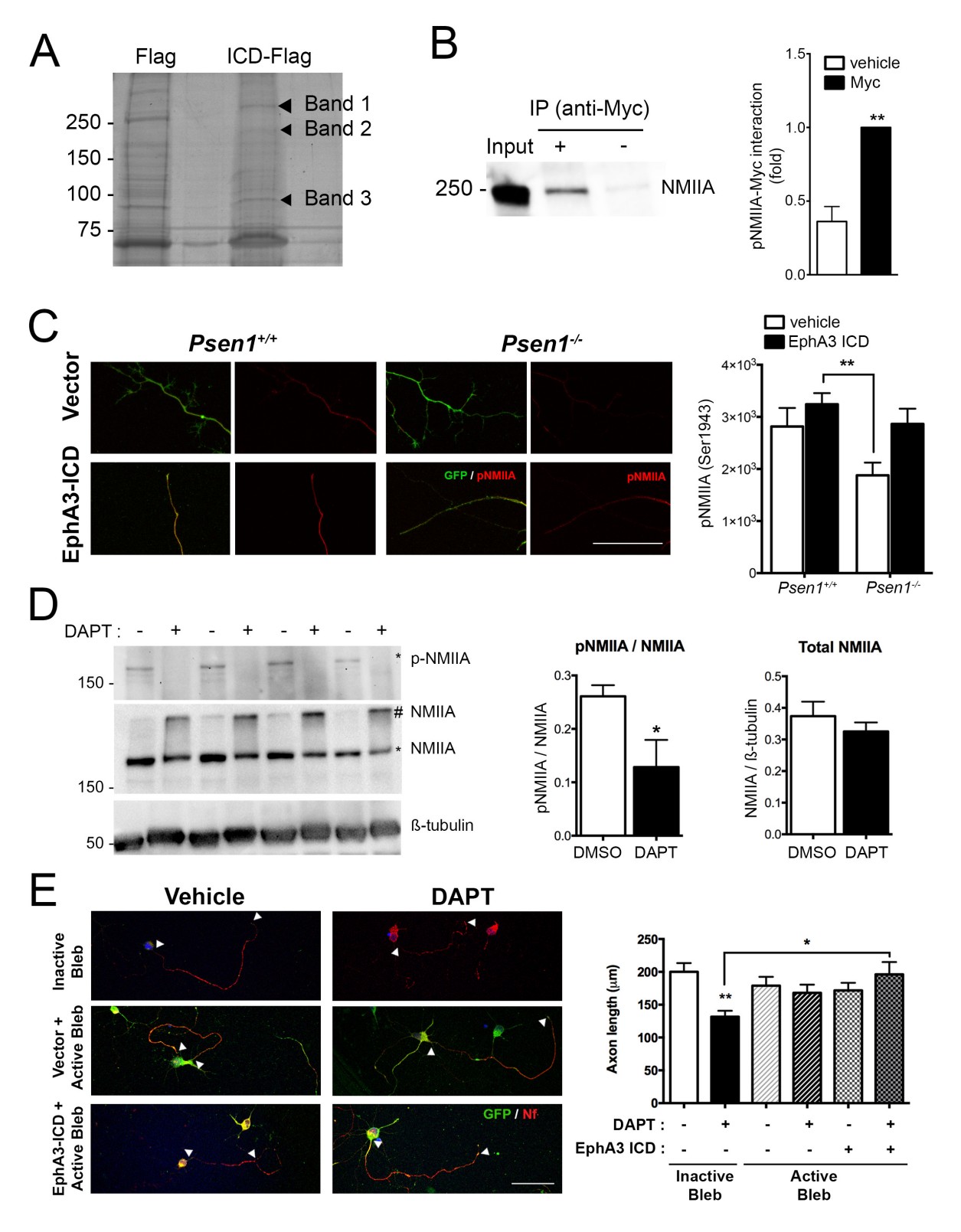

**Figure 6.** EphA3 ICD interacts with non-muscle myosin IIA and regulates its phosphorylation levels. (**A**) Proteomic analysis of EphA3 interacting proteins. HEK293 cells overexpressing vector (Flag) or EphA3 ICD-Flag (ICD-Flag) were immunoprecipitated with anti-Flag antibody, and proteins were resolved by SDS-PAGE and stained with Coomassie Blue. Specific proteins (Band 1, 2 and 3) were identified by MALDI-TOF/TOF mass spectrometry. Band one corresponds to NMIIA. (**B**) Binding of EphA3 ICD-myc to NMIIA/myosin 9. Coimmunoprecipitation (IP) of endogenous NMIIA and EphA3

*Figure 6 continued on next page*

*Figure 6 continued*

ICD-myc in HEK293 cells. Data are mean ± SEM of three experiments. Unpaired two-tailed Student's *t* test: **p<0.01. (C) Reduced pNMIIA in axons of *Psen1*$^{-/-}$ neurons. Immunofluorescence staining (left) and quantitative analysis (right) of phosphorylated NMIIA heavy chain (S1943) (pNMIIA: red) in axons of *Psen1*$^{+/+}$ and *Psen1*$^{-/-}$ neurons. *Psen1*-deficient neurons show reduced axonal pNMIIA intensity that is reversed by expressing EphA3 ICD (GFP, green) Scale bar, 50 µm. Data are mean ± SEM of 3 experiments (n = 25–33 neurons/group). Two-way ANOVA followed by Bonferroni *post hoc* test: **p<0.01. (D) Pharmacological PS/γ-secretase inhibition reduces phosphorylated NMIIA levels. DAPT treated hippocampal neurons show reduced pNMIIA (S1943) and soluble NMIIA (indicated with *), and increased insoluble/aggregated NMIIA (indicated with #). Multiple independent cultures are shown. Data are mean ± SD of four independent neuronal cultures. Unpaired two-tailed Student's *t* test: *p=0.05. (E) Pharmacological inhibition of NMII by blebbistatin rescues the defects in axon length in DAPT-treated hippocampal neurons. Immunofluorescence images of hippocampal neurons stained for GFP (EphA3-ICD or vector-positive neurons; green) and neurofilament (SMI312, red). Scale bar, 50 µm. Data are mean ± SEM of 3 experiments (n = 44–56 neurons/group). Two-way ANOVA indicates a significant DAPT treatment x ICD/blebbistatin interaction. *p<0.05, **p<0.01.
DOI: https://doi.org/10.7554/eLife.43646.010

The following figure supplements are available for figure 6:

**Figure supplement 1.** Reduced pNMIIA/F-actin colocalization in growth cones of *Psen1*$^{-/-}$ neurons.
DOI: https://doi.org/10.7554/eLife.43646.011
**Figure supplement 2.** NMIIA regulates axon retraction.
DOI: https://doi.org/10.7554/eLife.43646.012

*Epha3* silencing) inhibit axon growth suggesting that both mechanisms occur in cellular conditions. It is puzzling that not only deletion of the EphA3 C-terminal but also removal of the PDZ or LBD impair recovery of axon length in EphA3- and/or PS1/γ-secretase-deficient neurons, suggesting that the EphA3 structural conformation is important for mediating axon growth. Whereas the role of specific EphA3 domains on regulation of cytoskeleton assembly is still unclear, PS1/γ-secretase-dependent axon growth is independent of classical EphA3 signaling. Ephrin-A5 does not affect EphA3 processing nor cause additional defects on axon morphology in EphA3- and PS/γ-secretase-deficient neurons. In contrast, both ligand and neuronal activity regulate PS1/γ-secretase-dependent processing of EphB2 and EphA4 (*Litterst et al., 2007*; *Inoue et al., 2009*). We cannot rule out the possibility that metalloprotease-mediated ephrinA shedding or other intracellular EphA3 effectors mediate PS-dependent axon elongation (*Hattori et al., 2000*; *Janes et al., 2009*; *Mohamed et al., 2012*). It is also plausible that EphA3 ICD could mediate axon growth by acting through transcriptional mechanisms as described for APP and Notch ICDs (*Lleó and Saura, 2011*).

The PS/γ-secretase/EphA3-dependent mechanism mediating axon elongation involves RhoA signaling, a pathway implicated in growth cone collapse (*Noren and Pasquale, 2004*; *Hall and Lalli, 2010*). PS1/γ-secretase-deficient neurons show increased RhoA activity, whereas EphA3 ICD inhibits RhoA and reverses axon growth defects. By contrast, a constitutive active RhoA mutant causes the opposite effect, that is, prevents EphA3 ICD-induced recovery of axon growth. Importantly, proteomic analyses revealed that EphA3 ICD interacts with NMIIA heavy chain, a cytoskeleton protein that promotes neurite retraction (*Wylie and Chantler, 2003*; *Gallo, 2006*; *Kubo et al., 2008*). PS1/γ-secretase-deficient neurons show increased insoluble NMIIA and reduced NMIIA phosphorylation (Ser1943), a form that dissociates or prevents cytoskeleton assembly of myosin filaments (*Breckenridge et al., 2009*; *Dulyaninova and Bresnick, 2013*). Future investigations are needed to uncover the mechanism by which EphA3 ICD enhances NMIIA phosphorylation. It is possible that EphA3 ICD could affect activity and/or localization of protein kinase C and casein kinase II, the main kinases that phosphorylate NMIIA heavy chain (*Breckenridge et al., 2009*). Interestingly, TGF-β increases NMIIA Ser1943 phosphorylation during epithelial-mesenchymal transition (*Beach et al., 2011*). Since TGF-β plays essential roles during neuron specification and activates RhoA-dependent signaling another possibility is that EphA3 ICD could regulate TGF-β signaling affecting RhoA and NMIIA phosphorylation. Nonetheless, our results suggest that PS1/γ-secretase promotes disassembly or inhibits formation of stable filaments leading to F-actin-mediated axon extension. This idea is consistent with previous findings indicating that myosin inhibition induces extension of growth cones and filopodia by reducing retrograde F-actin flow (*Lin et al., 1996*). On the contrary, EphA3 ICD elevates phosphorylated NMIIA heavy chain and enhances NMIIA/actin colocalization in PS-deficient axons, indicating that it induces axon growth by promoting filament disassembly. Since pharmacological inhibition of NMIIA filament assembly reverses axon growth defects in PS/γ-secretase-deficient neurons, and it does not cause additional effects in the presence of EphA3 ICD, we conclude

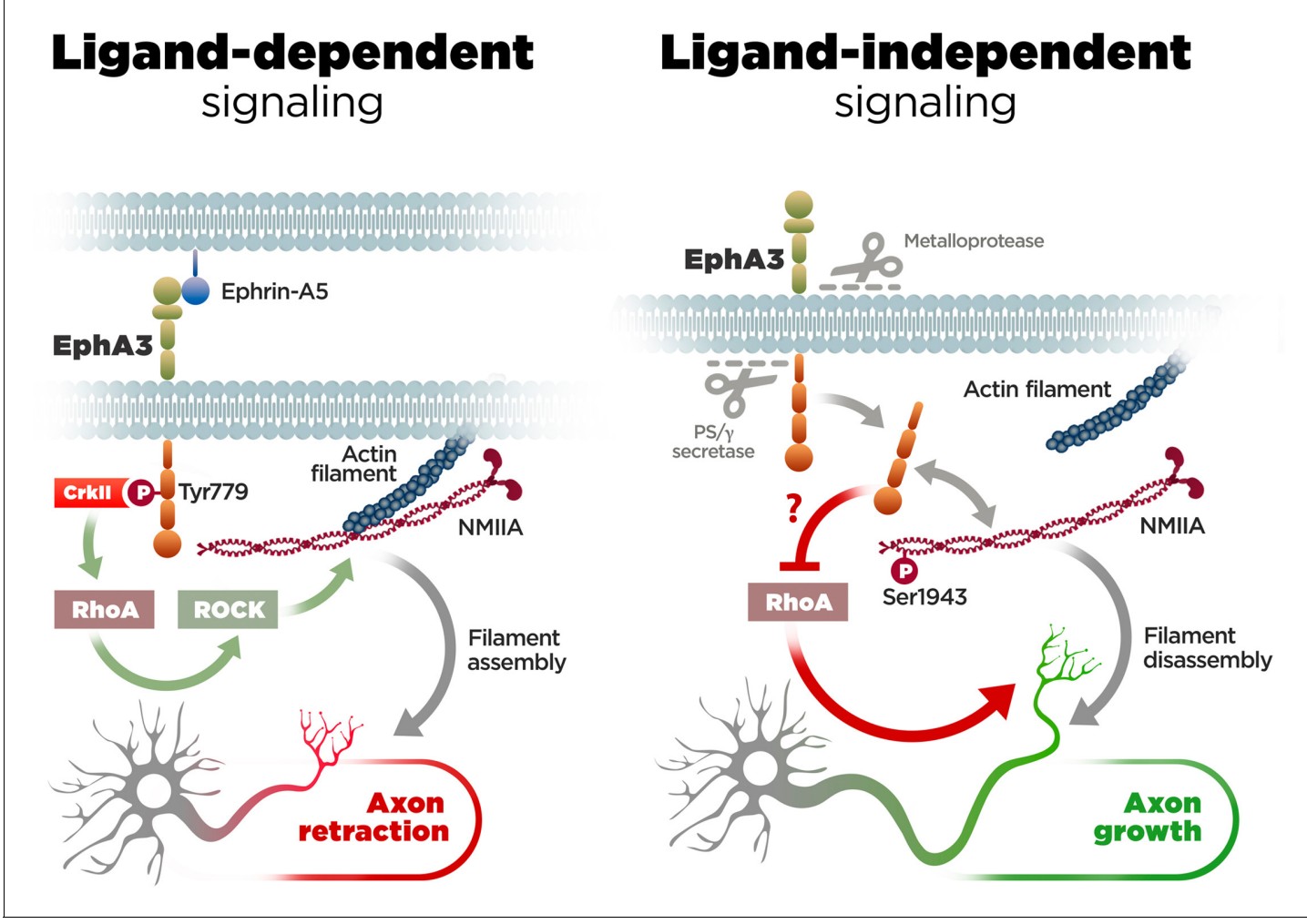

**Figure 7.** Molecular mechanisms of axon growth mediated by ephrin-dependent and -independent EphA3 signaling. Proposed model of ligand-dependent and -independent EphA3 signaling regulating axon growth in neurons. Left: Ephrin-A5 induces EphA3 phosphorylation (Tyr779), CrkII binding and activation of RhoA/ROCK, resulting in actin/NMIIA filament assembly and axon retraction in hippocampal neurons (*Figure 4*). Right: EphA3 ICD, generated by sequential cleavage of EphA3 by matrix metalloproteinase/ADAM and PS/γ-secretase proteases, promotes axon growth possibly by inhibiting RhoA signaling, and binding to and increasing phosphorylation (Ser1943) of NMIIA heavy chain in axons (*Figures 2*, *3*, *5* and *6*). Inactivation of PS/γ-secretase decreases phosphorylated NMIIA and NMIIA/actin colocalization in axons, which stabilizes membrane cytoskeleton-associated NMIIA resulting in assembly or blocking disassembly of filaments, and leads to inhibition of axon growth (*Figure 1*).

DOI: https://doi.org/10.7554/eLife.43646.013

that PS1/γ-secretase/EphA3 signaling mediates axon growth by promoting disassembly or preventing assembly of axonal NMIIA/actin filaments (*Figure 7*).

The knowledge of NMIIA function in the nervous system is still very limited. Our results may pave the way for future investigations on EphA3/NMIIA function in brain development and degeneration. Our findings are also relevant for memory-related diseases in which disrupted axon morphology occurs in some cerebral pathological conditions. Specifically, loss-of-function *Psen1* mutations cause dramatic changes in tau-related axon morphology and transport and synapses, which are early events in the pathogenesis of AD (*Stokin et al., 2005*; *Peethumnongsin et al., 2010*). PS/γ-secretase inactivation in neurons increases tau pathology and axonal transport deficits leading to synaptic plasticity and memory deficits and neurodegeneration (*Saura et al., 2004*; *Peethumnongsin et al., 2010*). Since EphA3/ephrin-A5 regulates synaptogenesis and septohippocampal projections in the limbic system (*Yue et al., 2002*; *Otal et al., 2006*), a brain circuit essential for learning, memory and emotional responses, our findings may be relevant for synaptic dysfunction in AD. Nonetheless, septohippocampal projections are severely damaged at early AD stages (*Riekkinen et al., 1987*),

whereas altered γ-secretase-dependent EphA4 processing was recently associated with synapse pathology in AD brains (*Matsui et al., 2012*). Since myosin NMIIB regulates synaptic actin dynamics during synaptic plasticity and memory (*Rex et al., 2010*), it will be interesting to study whether NMIIA regulates the function of PS1/γ-secretase on synapses. Considering the experimental therapies targeting Eph and myosin in nerve regeneration and cancer pathologies (*Boyd et al., 2014*), it is conceivable that a better understanding of the PS/γ-secretase/EphA3/NMIIA crosstalk may provide new perspectives on common mechanisms regulating human brain development and diseases.

# Materials and methods

**Key resources table**

| Reagent type (species) or resource | Designation | Source or reference | Identifiers | Additional information |
|---|---|---|---|---|
| Cell line (*H. sapiens*) | HEK293 (Human embryonic kidney 293 cells) | American Type Culture Collection | | |
| Cell line (*M. musculus*) | PS1/PS2$^{-/-}$ MEF (mouse embryonic fibroblast cells) | B. De Strooper (Katholieke Universiteit Leuven, Belgium) | | |
| Strain (*M. musculus*) | *Psen1$^{+/+}$*; *Psen1-/-*; *Psen2$^{-/-}$* | J. Shen (Brigham and Women Hospital, USA) | MGI: 1202717 | PMID: 9160754 |
| Antibody | anti-EphA3 C19 (rabbit polyclonal) | Santa Cruz Biotechnology | Cat. #: sc-919 RRID: AB_2099221 | WB (1:1000) |
| Antibody | anti-EphA3 H80 | Santa Cruz Biotechnology | Cat. #: sc-25456 RRID: AB_2099214 | ICC (1:200) IHC (1:200) |
| Antibody | anti-EphA3 L18 (rabbit polyclonal) | Santa Cruz Biotechnology | Cat. #: sc-920 RRID: AB_2099218 | WB (1:1000) |
| Antibody | anti-EphA3 (mouse monoclonal) | ThermoFisher Scientific | Cat. #: 5E11F2 RRID: AB_10104885 | WB (1:1000) IP (5 μg) |
| Antibody | anti-CrkII | BD Biosciences | Cat. #: 610035 RRID: AB_397451 | WB (1:5000) IP (5 μl) |
| Antibody | anti-Neurofilament; SMI312 (mouse monoclonal) | Covance | Cat. #: SMI-312R RRID: AB_2314906 | ICC (1:1000) |
| Antibody | anti-tau; TNT-1 (mouse monoclonal) | Merck Millipore | Cat. #: MAB3420 RRID: AB_94855 | IHC (1:400) |
| Antibody | anti-Nestin; rat-401 (mouse monoclonal) | Abcam | Cat. #: ab11306 RRID: AB_1640723 | IHC (1:200) |
| Antibody | anti-NeuN (mouse monoclonal) | Merck Millipore | Cat. #: MAB377 RRID: AB_2298772 | IHC (1:1000) |
| Antibody | anti-MAP2 (mouse monoclonal) | Sigma-Aldrich | Cat. #: M9942 RRID: AB_477256 | ICC (1:300) IHC (1:800) |
| Antibody | anti-Doublecortin (rabbit polyclonal) | Abcam | Cat. #: ab18723 RRID: AB_732011 | IHC (1:1000) |
| Antibody | PS1; APS11 (mouse monoclonal) | Abcam | Cat. #: ab15456 RRID: AB_301867 | ICC (1:200) IHC (1:200) |
| Antibody | PS1 loop (rabbit polyclonal) | Merck Millipore | Cat. #: AB5308 RRID: AB_91785 | WB (1:4000) |
| Antibody | APP C-terminal Saeko (rabbit polyclonal; serum) | M. Shoji (Gunma Univ School Medicine, Japan) | - | WB (1:7500) |
| Antibody | ß-actin [AC-15] (mouse monoclonal) | Sigma-Aldrich | Cat. #: A1978 RRID: AB_476692 | WB (1:60000) |
| Antibody | anti-Flag [FG4R] (mouse monoclonal) | Abcam | Cat. #: ab127420 RRID: AB_11157374 | IP (10 μl) |

*Continued on next page*

*Continued*

| Reagent type (species) or resource | Designation | Source or reference | Identifiers | Additional information |
|---|---|---|---|---|
| Antibody | anti-HA (mouse monoclonal) | Cell Signaling Technology | Cat. #: 2362 | WB (1:1000) |
| Antibody | AlexaFluor 594-phalloidin | ThermoFisher Scientific | Cat. #: A12381 RRID: AB_2315633 | ICC (1:50) |
| Antibody | GFP (chicken polyclonal) | Abcam | Cat. #: 13970 RRID: AB_371416 | ICC (1:1000) |
| Recombinant DNA reagent | EphA3-GFP | P. W. Janes (Monash University, Australia) | | |
| Recombinant DNA reagent | EphA3-HA | This study | | |
| Recombinant DNA reagent | EphA3-ΔICD | P. W. Janes (Monash University, Australia) | | |
| Recombinant DNA reagent | EphA3-ΔLBD | P. W. Janes (Monash University, Australia) | | |
| Recombinant DNA reagent | EphA3-ΔPDZ | P. W. Janes (Monash University, Australia) | | |
| Recombinant DNA reagent | EphA3-ICD | This study | | |
| Recombinant DNA reagent | EphA3-ICD-Flag | This study | | |
| Recombinant DNA reagent | EphA3-ICD-pCAGIG | This study | | |
| Recombinant DNA reagent | RhoA T19N | X. R. Bustelo (Centro Investigación del Cáncer, CSIC, Spain) | | |
| Recombinant DNA reagent | RhoA Q63L | P. Crespo (IBBTEC, Santander, Spain) | | |
| Recombinant DNA reagent | NMIIA WT | A. Bresnick (Albert Einstein College Medicine, NY, USA) | | |
| Recombinant DNA reagent | NMIIA S1943A | A. Bresnick (Albert Einstein College of Medicine, NY, USA) | | |
| Sequenced-based reagent | *Epha3* ShRNA | This study | | |
| Sequenced-based reagent | RT-qPCR primers | This study | | See Materials and methods |
| Peptide, recombinant protein | ephrin-A5 Fc chimera | R and D Systems | Cat. #: 374-EA | |
| Commercial assay or kit | RhoA G-LISA Activation Assay Kit | Cytoskeleton Inc | Cat. #: BK124 | |
| Commercial assay or kit | Rnease Mini kit | Qiagen | Cat. #: 74106 | |
| Chemical compound, drug | DAPT | Sigma-Aldrich | Cat. #: D5942 | 5 µM |
| Chemical compound, drug | L685,458 | Tocris | Cat. #: 2627 | 5 µM |
| Chemical compound, drug | GM6001 | Enzo | Cat. #: BML-EI300-0001 | 25 µM |
| Chemical compound, drug | 1,10-PNT | Sigma | Cat. #: 131377 | 83 µM |

*Continued on next page*

*Continued*

| Reagent type (species) or resource | Designation | Source or reference | Identifiers | Additional information |
|---|---|---|---|---|
| Chemical compound, drug | MMP9/13 | Calbiochem | Cat. #: 444252 | 10 µM |
| Chemical compound, drug | Y27632; ROCK-1 inhibitor | Calbiochem | Cat. #: 688000 | 10 µM |
| Software, algorithm | GraphPad Prism | GraphPad Prism (https://www.graphpad.com/) | RRID:SCR_015807 | Version 6 |
| Software, algorithm | ImageJ | ImageJ (https://imagej.nih.gov/ij/) | RRID:SCR_003070 | |
| Software, algorithm | Imaris | Imaris (https://imaris.oxinst.com/) | RRID:SCR_007370 | Version 8 |

## DNA and viral constructs

Human EPHA3 (EphA3), -GFP tagged and cytoplasmic-truncated ΔICD, ΔLBD and ΔPDZ were generated and cloned in BstX1/NotI sites of the pEFBos vector as previously described *Janes et al. (2005)*. To generate EphA3 C-terminal hemagglutinin (HA) tag, the EphA3-GFP plasmid was digested with XmaI and BamHI to remove GFP and ligated to complementary 5′-XmaI/3′-BamHI (underlined) oligonucleotides containing a hemagglutinin (HA) sequence (in bold): 5′-CCGGGGG TGGTGGATCCTACCCTTACGACGTTCCTGATTACGCTAGCCTCGAATTCTAATAG-3′ and 5′-GA TCCCTATTAGAATTCGAGGCTAGCGTAATCAGGAACGTCGTAAGGGTAGGATCCACCACCC-3′.

Bioinformatic prediction of EphA3 ICD was obtained by protein sequence alignment of EPHA3 and known PS1/γ-secretase substrates (EPHB2, APLP1, NRXN1, CADH1, PVRL1, NOTCH1 and CD44) using ClustalW2-EMBL (http://www.ebi.ac.uk). Theoretical molecular weight was obtained with ExPASy and Protein Molecular Weight Bioinformatic tools. The EphA3-ICD fragment (aa 561–983) was generated from hEPHA3 by PCR cloning with the following forward and reverse primers (underlined 3′EcoRI and 5′KpnI recognition sequences; human EphA3 sequence in bold): 5′-CCGGAA TTCCCGTATGTTTTGATTGGGAGGTTCTGTGG-3′ and 5′-CGGGGTACCCCGTTACACGGGAAC TGGGCCAT-3′ inserted in pCMV-HA (N-terminal; Clontech PT3283-5) and cloned into EcoRI/ApaI pcDNA3-3xFLAG vector (ThermoFisher Scientific,Whaltham, MA, USA). For in vivo in utero electroporation, EphA3 ICD was subcloned into a pCAGIG vector containing the internal ribosomal entry site (IRES)-GFP cassette. Complementary oligonucleotides for mouse *Epha3* shRNA were as follows: Sh-*Epha3* forward: 5′-gatccccGGGAATGCTCCGTGGGATAttcaagagaTATCCCACGGAGCA-TTCCCtttt-3′; Sh-*Epha3* reverse: 5′-agctaaaaaGGGAATGCTCCGTGGGATAtctcttg- aaTA TCCCACGGAGCATTCCCggg-3′. The scramble control oligonucleotides used were as follows: forward, 5′-gatcCCCGGAGAGCGTAGCGACTGTTttcaagagaAACA- GTCGCTACGCTCTCCtttt-3′ and reverse, 5′-agctaaaaaGGAGAGCGTAGCGACTG- TTtctcttgaaAACAGTCGCTACGCTCTCCGGG-3′. Oligonucleotides were cloned into BglII/HindIII sites of the pSUPER.retro vector. Lentiviral vectors were obtained by digesting EcoRI-ClaI sites from pSUPER-shRNA and the resulting insert was subcloned into pLVTHM lentiviral vector. Lentiviral particles were generated in HEK293T cells transfected with pLVTHM-Sh, pSPAX2, and pM2G vectors. Dominant negative mutant RhoA T19N was cloned in pCEFL-AU5 (gift of X. Bustelo, CSIC, Universidad de Salamanca, Spain) and RhoA Q63L cloned in pCEFL-HA was from P. Crespo (IBBTEC, Santander, Spain). NMIIA WT and S1943A, cloned into HindIII and SAlI sites of pEGFP-C3, were previously described (*Dulyaninova et al., 2007*).

The following antibodies were used: rabbit anti-EphA3 C19 (sc-919), H80 (sc-25456) and L-18 (sc-920) from Santa Cruz Biotechnology (Dallas, TX, USA); mouse anti-EphA3 (5E11F2, ThermoFisher Scientific); CrkII (BD Biosciences, Franklin Lakes, NJ, USA); neurofilament (SMI312), doublecortin, PS1 (APS11) and anti-Flag (FG4R) from Abcam (Cambridge, UK); tau (TNT-1), nestin (rat-401), NeuN and PS1 loop from Merck Millipore (Burlington, Massachusetts, USA); MAP2 and β-actin from Sigma-Aldrich (Saint Louis, MO, USA); anti-HA (#2362, Cell Signaling Technology Danvers, MA, USA) and AlexaFluor 594-phalloidin (ThermoFisher Scientific).

## Quantitative Real-Time RT-PCR

Mouse hippocampal neurons were cultured for 7 DIV and RNA was isolated using RNeasy Mini Kit (Qiagen) according to the manufacturer's instructions. Purified RNA (500 ng) was reverse-transcribed and amplified using Power SYBR Green PCR Master Mix (Cat. #4367659; ThermoFisher Scientific) in a 7500 Fast System (Applied Biosystems, Waltham, MA, USA). Data analysis was performed by the comparative Ct method using the Ct values and the average value of PCR efficiencies obtained from LinRegPCR software. Gene expression was normalized to *Gapdh and hypoxanthine guanine phosphoribosyl transferase (Hprt1).* Primers used were as follows: *Epha1* forward, 5'- CACCAGTTTCCAGAAGCCTG-3', reverse, 5'-CATAAATCCCGATCAGCAGAGC-3'; *Epha2* forward, 5'-TCCAAG TCAGAACAACTAAAGC-3', reverse, 5'-GGTCTTCGTAAGTGTGAGGA-3'; *Epha3* forward, 5'-C TAGCCCAGACTCTTTCTCC-3'; reverse, 5'-CGGAAATAGCAATCATCACCA-3'; *Epha4* forward, 5'-GAGAGTTCCAGACCAAACAC, reverse, 5'-ACTACAGCAGAGAATTCAGGG-3'; *Epha5* forward, 5'-TCCGCACACTTATGAAGATCC-3', reverse, 5'-TCACCAAATTCACCTGCTCC-3'; *Epha6* forward, 5'-TGATCCAGACACCTATGAAGAC-3', reverse, 5'-CAAATTCACCTGCTCCAATCAC-3'; *Epha7* forward, 5'-GCATTTCTCAGGAAACACGA-3', reverse, 5'-ACCTCTCAACATTCCTACCA-3'; *Epha8* forward, 5'-TCTAGCCTATGGTGAACGAC-3', reverse, 5'-CTGATGACATCCTGGTTGGT-3'; *Epha10* forward, 5'-TCCTGAGACTCTACAGTTTGG-3', reverse, 5'-GCCTTGATTACATCTTGTCCAG-3'.

## Cell lines and biochemical analyses

We used HEK 293 T cells (source: American Type Culture Collection) and immortalized PS1/PS2 mouse embryonic fibroblasts (provided by B. de Strooper, KU Leuven, Belgium) free of mycoplasma and authenticated using Short Tandem Repeat and PCR standard methods. Cells were lysed in lysis buffer (50 mM Tris HCl, pH 7.4, 150 mM NaCl, 2 mM EDTA, 1% NP40, 1 mM PMSF) containing protease and phosphatase inhibitors (Roche, Basel, Switzerland). Lysates were pre-cleared by centrifugation (10,000 x *g*, 10 min, 4°C). Proteins were quantified with the BCA protein assay kit (ThermoFisher Scientific), resolved on 8–12.5% SDS-polyacrylamide gel electrophoresis and detected by Western blotting (*España et al., 2010*). For co-immunoprecipitation, cells were washed in PBS and lysed in cold immunoprecipitation buffer (50 mM Tris HCl, pH 7.4, 150 mM NaCl, 2 mM EDTA, 1% NP40, 1 mM PMSF, phosphatase and protease inhibitors). Fresh cell lysates (300–500 μg protein) were precleared with Protein G and incubated with PS1 antibody (APS11; Abcam) and Protein G (ThermoFisher Scientific) at 4°C before extensive washes and analysis by Western blotting as described (*Saura et al., 1999*). For brain co-immunoprecipitation, cortex was homogenized in cold Tris 50 mM buffer (pH 7.4), and nucleus and tissue debris were discarded (1000 x *g*, 10 min, 4°C). Membrane extracts were centrifuged twice (12,000 x *g*, 30 min, 4°C) and resuspended in RIPA buffer (50 mM Tris-HCl pH 7.4, 100 mM NaCl, 1% Triton-X100, 0.5% sodium deoxycholate, 0.2% SDS, 1 mM EDTA and phosphatase and protease inhibitors). Supernatants were incubated overnight with PS1 NT antibody (#529591; Merck Millipore) and Dynabeads protein G (ThermoFisher Scientific). For ectoshedding studies, conditioned media were recollected after 48 hr of EphA3 transfection. The protease inhibitor PMSF (100X) was added upon conditioned media before cells were removed by centrifugation (1,000 rpm, 5 min, room temperature). Then, proteins in conditioned media were concentrated using Amicon Ultra 10K filters (Merck Millipore), resolved on 10% SDS-polyacrylamide gel electrophoresis and analysed by Western blotting with 5E11F2 and L18 antibodies.

## Neuronal culture and pharmacological treatments

*Psen1*[+/+], *Psen1*[-/-] and *Psen2*[-/-] mouse embryos (C57/BL6 background) were obtained from *Psen1*[+/-] x *Psen1*[+/-] or *Psen2*[+/-] x *Psen2*[+/-] crossings as described (*Shen et al., 1997*). Mouse hippocampal neurons (E15.5) were cultured for four days in vitro (DIV) at a density of $1.6 \cdot 10^4$ cells/cm$^2$ or $4.5 \cdot 10^4$ cells/cm$^2$ in poly-D-lysine coated 24-well or 60 mm dishes, respectively (*España et al., 2010*) and. Neurons were treated with DAPT (250 nM; Sigma-Aldrich), GM6001 (4 μM; Merck Millipore) or transduced with ShRNA lentiviral vectors at 0 DIV, and/or transfected at 2 DIV with EphA3 cDNAs, RhoA DN, RhoA CA and/or EGFP plasmids with LipofectAMINE 2000 (ThermoFisher Scientific) or treated with Y27632 (10 μM; Merck Millipore). For blebbistatin experiments, hippocampal neurons were transduced with EphA3-ICD lentiviral vectors at 1 DIV, treated at 2 DIV with active or inactive blebbistatin (20 μM) and DAPT (250 nM; Sigma-Aldrich). Neurons were transfected with NMIIA WT or

S1943A at 2 DIV, treated with DAPT (250 nM; Sigma-Aldrich). For ephrin-A5 activation, human ephrin-A5 Fc chimera (R and D Systems, Minneapolis,MN, USA) clusterized (1:10 molar ratio) with anti-human IgG Fc (Jackson ImmunoResearch, Cambridge, UK) was incubated at 0.18 ng/mm$^2$ as described (*Lawrenson et al., 2002*). HEK 293 T cells (5·10$^4$ cells/cm$^2$) were cultured in DMEM (Sigma) supplemented with 10% fetal bovine serum (ThermoFisher Scientific) and transfected with LipofectAMINE 2000. Cells were treated with DAPT (5 µM; Sigma-Aldrich), L685,458 (R and D Systems) or metalloprotease inhibitors GM6001 (25 µM; Merck Millipore), 1,10-phenanthroline (50 µM; Sigma-Aldrich) or MMP9/13 (10 µM; Merck Millipore). Animal experimental procedures were conducted following the European Union guidelines for animal care and use (2010/63/EU) according to the approved Animal and Human Ethical Committee (CEEAH) protocol (CEEAH 2896; DMAH 8787) of the Universitat Autònoma de Barcelona.

## γ-secretase and RhoA activity assays

In vitro γ-secretase assay was performed as described (*Sastre et al., 2001*). Briefly, EphA3- transfected HEK293 cells or embryonic brains were lysed in hypotonic buffer (10 mM MOPS, pH 7.0, 10 mM KCl, protease inhibitors). Samples were centrifuged (1000 x *g*, 15 min, 4°C) and the postnuclear supernatant was recentrifuged (16,000 x *g*, 20 min). Pellets were resuspended in assay buffer (150 mM sodium citrate, pH 6.4) containing vehicle or DAPT at 37°C for the indicated time. Supernatant (S100) and pellet (P100) were collected after centrifugation (100,000 x *g*, 1 hr) and analysed by SDS-PAGE and immunoblotting with C19 or 5E11F2 antibodies. For RhoA activity assays, 4 DIV neurons cultured in the presence of vehicle or DAPT and treated with ephrin-A5 for the indicated time or SK-N-AS cells transfected with EphA3-ICD for 48 hr were incubated with vehicle or lysophosphatidic acid (10 µM; Sigma-Aldrich) for 5 min. RhoA activity was measured in freshly prepared cell lysates using the RhoA G-LISA Activation Assay Kit (Cytoskeleton Inc, Denver,CO, USA).

## Immunohistochemical and DiI staining

Mouse embryos (E15.5) were perfused intracardially with 4% formaldehyde/PBS solution prior to paraffin embedding. Coronal brain sections (5 µm) were deparaffinized in xylene, rehydrated and microwave-heated with sodium citrate (10 mM; Sigma) for antigen retrieval (*Saura et al., 2004*). Sections were incubated with antibodies against neurofilament (SMI312; 1:500), tau (1:400), nestin (1:200), doublecortin (1:1,000) and secondary AlexaFluor-488/555-conjugated goat IgGs (1:400) and Hoechst (1:10,000; ThermoFisher Scientific). Images were obtained with a Zeiss LSM700 laser scanning microscope (20x). For axonal quantification in tissue, a 3D reconstruction picture of ventricular zone and cortical plate was generated from multiple stacks of each section. Axons were semi automatically tracked by using Filament Tracer (Imaris, Bitplane Inc).

For DiI tracing, embryonic brains at E13.5 were dissected and fixed overnight with 4% PFA in PBS. Then, a small crystal of DiI probe [1,1'-Dioctadecyl-3,3,3',3'-Tetramethylindocarbocyanine Perchlorate ('DiI'; DiIC18(3)] (ThermoFisher Scientific, D282) was placed in the dorsal part of the somatosensory region of the cortex and the brains were incubated in 4% PFA/PBS for 10 days at 37°C to let the probe to diffuse. Brains were embedded in 1% agarose/PBS and sectioned (80 µm thickness) with a Vibratome (Leica Microsystems, Wetzlar, Germany). Sections were blocked with 5% BSA in PBS and immunostained with an anti-neurofilament antibody (2H3, Developmental Studies Hybridoma Bank, Iowa City, IA, USA) and the appropriate secondary antibody coupled to Cy2 (Jackson ImmunoResearch). Samples were counterstained with DAPI for nuclear staining. Finally, sections were collected on slides with PBS/1% glycerol and visualized with a confocal microscope (Olympus FV1000, Shinjuku, Japan). Axonal length measurements were done with FilamentTracer tool (Auto-Path method) from Imaris 8.1 Software (Bitplane). Axons were tracked following DiI (in red) and neurofilament (in green) stainings.

## Axon length imaging analysis

Hippocampal neurons (4 DIV) were fixed with 4% paraformaldehyde in PBS, permeabilized with 0.02% saponin and blocked with 10 mM glycine and 5% BSA in TBS. Staining was performed with mouse anti-SMI312 (1:500; Abcam), chicken anti-GFP (1:1000; Abcam) and/or phalloidin-Alexa594 (1:40; ThermoFisher Scientific) and detected with anti-mouse AlexaFluor488 or 594 secondary antibodies (1:400; ThermoFisher Scientific) and Hoechst 33258. Images were captured with a Nikon i90

fluorescence microscope and analyzed with Image J (NIH, USA) and Metamorph (Molecular Devices, Sunnyvale, CA, USA) softwares. Axons were manually tracked by following the neurofilament staining from the cell body to the actin neurite tip or GFP staining. Axon length was obtained from at least three independent experiments counting three-six independent coverslips (n = 30 neurons/coverslip) per condition. For axonal length imaging using NMIIA mutants, images were captured with a Zeiss LSM700 laser scanning microscope and analyzed with FilamentTracer tool from Imaris 8.1 Software (Bitplane AG, Zurich, Switzerland).

## Colocalization analysis

Hippocampal neurons (4 DIV) were fixed with 4% paraformaldehyde in PBS, permeabilized with 0.02% saponin and blocked with 10 mM glycine and 5% BSA in TBS. Neurons were stained with monoclonal PS1 (APS11; 1:200) and rabbit polyclonal EphA3 (H-80; 1:200) antibodies. Image stacks (0.5 µm; 20 stacks/condition) were obtained with a Zeiss LSM700 laser scanning microscope. Colocalization imaging quantification was performed by analysing pixel by pixel each channel in 3 to 5 regions of interest in the central slices of each stack by using the colocalization highlighter plugin of ImageJ software (Institute Jacques Monod, Service Imagerie). Results were expressed as percentage of colocalized pixels relative to total pixels of each channel.

## In utero electroporation assays

Pregnant *Psen1*[+/-] females (E14.5) from *Psen1*[+/-] x *Psen1*[+/-] intercrosses were deeply anesthetized during the whole experimental procedure. Females were administrated intraperitoneally with the muscle relaxant Ritodrine (Sigma-Aldrich) and subcutaneously with the analgesic buprenorphine (Buprex). After abdominal laparotomy the uterine horns were carefully exposed and lubricated continuously with saline (NaCl 0.9%). Two to four µg (2–4 µl) of purified control (mCherry) or EphA3ICD pCAGIG-IRES-GFP plasmids diluted in PBS containing 0.025% Fast Green (Sigma-Aldrich) were injected through the uterine wall in the brain lateral ventricle using a glass capillary sharpened with a glass puller (P-97, Sutter Instrument, Novato, CA, USA). Platinum electrodes were placed around the head of the injected embryos and five electric pulses (30 mV and 50 ms, each) with 950 ms interval were applied with and electroporation system. After electroporation, the uterine horns were placed back in the abdominal cavity and the abdomen of the pregnant female was sutured. Twenty-four hours later, the embryos were collected and the brains were dissected and fixed for 4 hr in 4% PFA/PBS, washed, cryoprotected with 30% sucrose/PBS and embedded in tissue freezing medium. Sections (20 µm) were obtained in a cryostat (Leica) and collected on Superfrost Plus slides (Thermo-Fisher Scientific). Slides were washed with PBS, permeabilized with PBS+0.1% Triton X-100 (PBT), block with 5% BSA and 5% donkey serum in PBT and immunostained with an anti-GFP antibody (600-101-215; Rockland Immunochemicals, Limerick, PA, USA) and the appropriate secondary antibody coupled to Cy2 (Jackson ImmunoResearch). Samples were counterstained with DAPI for nuclear staining. Finally, samples were visualized with a confocal microscope (Olympus FV1000). Axonal length analyses of GFP labeling were performed with FilamentTracer tool (AutoPath method) from Imaris 8.1 Software (Bitplane).

## Morphological analysis of growth cones

Hippocampal neurons (4 DIV) were fixed with 4% paraformaldehyde in PBS, permeabilized with 0.02% saponin and blocked with 10 mM glycine and 5% BSA in TBS. Staining was performed with rabbit polyclonal pNMIIA (AB2974; 1:100), chicken polyclonal anti-GFP (1:1000; Abcam) and phalloidin-Alexa594 (1:50; ThermoFisher Scientific) diluted in 1% normal goat serum in TBS, and detected with anti-mouse AlexaFluor488 or 594 secondary antibodies (1:300; ThermoFisher Scientific) and Hoechst 33258. Images were captured with a Zeiss LSM700 laser scanning microscope. Axon growth cones were manually classified as collapsed, those that have no lamellipodia and no more than two filopodia or as non-collapsed (the remaining). Quantitative immunofluorescence of pNMIIA of the axon was measured by ImageJ software (Institute Jacques Monod, Service Imagerie, France) and expressed as area integrated intensity. Colocalization between pNMIIA and F-actin was determined using the tool ImarisColoc from Imaris 8.1 Software (Bitplane). Colocalization spots are indicated in pink (pNMIIA vs F-actin spots) and purple (F-actin vs pNMIIA spots). Results are expressed as percentage of colocalized spots (pink and purple) relative to total spots.

## Proteomic approaches

PS/γ-secretase-mediated cleavage site in EphA3 was identified by LC-MS/MS analysis. In addition, the cleaving site was also simultaneously predicted by sequence alignment of EPHA3 and well-established PS1/γ-secretase substrates (EPHB2, APLP1, NRXN1, CADH1, PVRL1, NOTCH1 and CD44) using ClustalW2-EMBL (http://www.ebi.ac.uk). Briefly, an in vitro γ-secretase assay in EphA3 overexpressing HEK2993 cells was performed as described in the Materials and methods section. Soluble (S100) samples were resolved on 8.5% SDS-PAGE and stained using silver staining. Bands were excised and in-gel digested with trypsin automatically (DigestPro MS; Bioanalytical Instruments, New York, NY, USA), and peptides were extracted in MeOH/H2O (2:1), 0.1% TFA. Samples were evaporated and reconstituted in 5 µl of 5% MeOH, 0.5% TFA and loaded into the chromatographic system. The LTQ XL Orbitrap was operated in the positive ion mode with a spray voltage of 1.8 kV. The spectrometric analysis was performed in a data dependent mode, acquiring a full scan followed by 5 LC-MS/MS scans of the five most intense signals from an inclusion list. The inclusion list included all the theoretical peptides generated after MS-digestion (ProteinProspector). The obtained mass spectrometry spectra of the identified **VLIGR** peptide is shown in *Supplementary file 1*.

We used a well established protocol to identify EphA3-ICD interacting proteins (*Free et al., 2009*). Briefly, Flag-ICD-transfected HEK293 cells were incubated with EBSS supplemented with 5 mM EDTA for 10 min at 37°C and 5% $CO_2$ and centrifuged at 200 x g, 10 min. Cells were lysed in low-detergent buffer (50 mM Tris HCl, pH 7.4, 150 mM NaCl, 2 mM EDTA, 1% NP40, 1 mM PMSF, protease and phosphatase inhibitors). Samples were solubilized for 1 hr at 37°C in an orbital shaker and centrifuged (20,000 x g, 40 min, 4°C) and supernatant was used for the next steps. Supernatants were pre-cleared with Protein-G for 3 hr on a rocking platform and incubated with Flag antibody (FG4R; Abcam) and Protein G (ThermoFisher Scientific) before extensive washes. Immunoprecipitated proteins were denatured at 95°C for 5 min, resolved on 8.5% SDS-PAGE and stained by Colloidal Coomassie reagent. Samples were in-gel digested with trypsin. Resulting peptides were analyzed by an Ultraflex MALDI-TOF/TOF mass spectrometer (Bruker, Billerica, MA, USA) and analysed by the database search (Mascot, Matrix Science) with a significance threshold of the MOWSE score of p<0.05 and then manually validated. SwissProt database restricted to *Homo sapiens* taxonomy was used for peptide identification.

## Statistical analysis

Statistical analysis was performed essentially by one- or two-way analysis of variance (ANOVA) followed by Bonferroni *post hoc* test or Student's *t* test using GraphPad Prism (GraphPad Software, La Jolla, USA). Values identified as outliers by Grubbs' test were not included in the analysis (p<0.05). Each experiment was performed using, at least, three independent experiments. When differences among groups did not reach statistical significance, sample size was calculated by a power analysis test. Values represent mean ± SD or SEM. Value differences were considered significant when p<0.05.

## Acknowledgements

The authors thank J Shen (Harvard Medical School, Boston, USA) for providing the *Psen1*[-/-] mice, B De Strooper (Katholieke Universiteit Leuven, Belgium) for providing the *Psen1*[+/+]/2[+/+] and *Psen1*[-/-]/2[-/-] mouse embryonic fibroblasts, A Bresnick (Albert Einstein College of Medicine, NY, USA) for kindly providing the NMIIA constructs, X R Bustelo (Centro Investigación del Cáncer, CSIC, Spain) for RhoT19N plasmid and P Crespo (IBBTEC, Santander, Spain) for RhoA Q63L plasmid. We are grateful to Silvia Ginés for critical review of the manuscript, and M Sastre for technical advise on the γ-secretase assays. We thank Núria Barba and Mar Castillo from the Servei de Microscopia i Histologia Units, Institut de Neurociències and Marta Cosín and Gisela Esquerda for technical assistance. We thank the animal house facilities of Universitat de Lleida and Universitat Autònoma de Barcelona for their expert technical assistance. We thank the UAB Proteomic facility and LP-CSIC/UAB (PT17/0019/0008), a member of ProteoRed network, for proteomic analyses. This study was funded by grants from the Ministerio de Economía y Competitivad (SAF2013-43900-R, SAF2016-80027-R and CIBERNED CB06/05/0042) and Generalitat de Cataluña (ICREA Academia and 2017 SGR749).

## Additional information

### Funding

| Funder | Grant reference number | Author |
| --- | --- | --- |
| Ministerio de Economía y Competitividad | SAF2013-43900-R | Carlos A Saura |
| Ministerio de Economía y Competitividad | SAF2016-80027-R | Carlos A Saura |
| Ministerio de Economía y Competitividad | CIBERNED CB06/05/0042 | Carlos A Saura |
| Generalitat de Catalunya | ICREA Academia | Carlos A Saura |
| Generalitat de Catalunya | 2017 SGR749 | Carlos A Saura |

The funders had no role in study design, data collection and interpretation, or the decision to submit the work for publication.

### Author contributions

Míriam Javier-Torrent, Conceptualization, Data curation, Formal analysis, Validation, Investigation, Methodology, Writing—original draft; Sergi Marco, Conceptualization, Data curation, Formal analysis, Supervision, Validation, Investigation, Methodology; Daniel Rocandio, Resources, Data curation, Formal analysis, Methodology; Maria Pons-Vizcarra, Formal analysis, Methodology; Peter W Janes, Conceptualization, Resources, Supervision, Methodology; Martin Lackmann, Conceptualization, Resources, Methodology, Writing—original draft; Joaquim Egea, Conceptualization, Resources, Supervision, Investigation; Carlos A Saura, Conceptualization, Supervision, Funding acquisition, Investigation, Writing—original draft, Writing—review and editing

### Author ORCIDs

Míriam Javier-Torrent (iD) https://orcid.org/0000-0002-6437-2829
Maria Pons-Vizcarra (iD) http://orcid.org/0000-0001-5008-5741
Carlos A Saura (iD) https://orcid.org/0000-0003-3692-5657

### Ethics

Animal experimentation: Animal experimental procedures were conducted following the European Union guidelines for animal care and use (2010/63/EU) according to the approved Animal and Human Ethical Committee (CEEAH) protocol (CEEAH 2896; DMAH 8787) of the Universitat Autònoma de Barcelona.

### Decision letter and Author response

Decision letter https://doi.org/10.7554/eLife.43646.018
Author response https://doi.org/10.7554/eLife.43646.019

## Additional files

### Supplementary files

• Supplementary file 1. Proteomic analysis of the PS/γ-secretase-mediated EphA3 cleavage site. Mass spectrometry spectra of the trypsin-digested band (~47–49 kDa) (top spectra) and specific mass spectrometry spectra obtained for peptide VLIGR showing the mass/charge (m/z) values (bottom spectra). Detected signals corresponding with theoretical ions are labeled with red asterisks.
DOI: https://doi.org/10.7554/eLife.43646.014

• Supplementary file 2. Identified peptides from NMIIA protein.
DOI: https://doi.org/10.7554/eLife.43646.015

• Transparent reporting form
DOI: https://doi.org/10.7554/eLife.43646.016

## Data availability

All data generated or analysed during this study are included in the manuscript and supporting files.

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
