## [Decision Letter]

[Editors’ note: this article was originally rejected after discussions between the reviewers, but the authors were invited to resubmit after an appeal against the decision.]

Thank you for submitting your work entitled "Presenilin/γ-secretase-dependent EphA3 processing mediates axon elongation through non-muscle myosin IIA" for consideration by *eLife*. Your article has been reviewed by three peer reviewers and has been overseen by a Reviewing Editor and a Senior Editor. The reviewers have opted to remain anonymous.

While the reviewers found the work interesting, the number of substantive questions raised was such that we feel we must reject it. In particular, the reviewers felt that the in vivo data relating to the specific role of EphA3 ICD in axon growth is lacking. See the comments (below) that provide more details with respect to this issue as well as other concerns raised by reviewers.

Addressing these issues would likely require more than the typical time allowed for a standard *eLife* revision. We would be happy to consider the manuscript at a more distant date, should you consider to pursue the work along the suggestions of reviewers.

*Reviewer #1:*

The authors report that the receptor EphA3 is cleaved by presenilin-1 in a ligand- (ephrinA) independent manner and that, the resulting intracellular domain promotes axon growth by phosphorylating non-muscle myosin IIA. Presenilin-1 was previously shown to interact with, and to cleave another axon guidance player, namely the netrin receptor DCC; a process that increases the fidelity of commissural axons pathfinding (Bai et al., 2011). Hence, this manuscript adds to our understanding of the mechanisms by which presenilin-1 regulates directional growth of axons.

Technically, the study is well conducted, the results are convincing, and the conclusions are supported by the findings. However, in vivo evidence is lacking as most experiments were performed in vitro. Authors may at least discuss better and stress axon defect similarities, overlaps, and differences between presenilins and EphA3/ephrinAs mutant mice.

*Reviewer #2:*

The present manuscript shows the potential role of presenilin/γ-secretase cleavage on axon elongation, indicating that the mechanism mediating this effect is the cleavage of EphA3/ephrin. The study demonstrates that the formation of the intracellular domain (ICD) seems to inhibit Rho signalling and interact with non-muscle myosin IIA (NMIIA), affecting filament assembly. The findings are new and interesting and the study is very well designed, however some points need to be further clarified:

1) The authors have only demonstrated the role of PS1 using PS1 knockout cells, although DAPT could also affect PS2 activity. Perhaps it would be interesting to use PS2 knockout cells or cells transfected with siRNA for PS2.

2) The study has been done in embryonic brains, are there also differences in axonal length/white matter density in older PS1 knockout mice?

3) Although this study points to the interaction of the ICD with Rho signalling and NMIIA, it should also be ruled out that the effect is not mediated by transcriptional related mechanisms, as it occurs with other intracellular fragments resulting from PS1 cleavage, such as NICD and AICD.

4) In Figure 5A-C the use of a Rho-kinase inhibitor does not seem to have any effect on axonal length in PS1 wild-type cells, by looking at the quantification shown in the graphs. Is this the case? What could be the interpretation? The sentence in subsection “EphA3 cleavage mediates axon elongation by inhibiting RhoA signalling” is a bit confusing and could be changed to "dominant-negative RhoA mutant (RhoA T19N) efficiently reversed axon length defects in PS1- and γ-secretase activity-deficient neurons".

5) The authors should propose a potential mechanism whereby the ICD affects NMIIA phosphorylation.

*Reviewer #3:*

The manuscript by Javier-Torrent and coworkers proposes a new mechanism by which presenilin-1/γ-secretase regulates axon growth. PS1 induces EphA3 cleavage generating an intracellular domain (ICD) fragment that is sufficient to promote axon growth, inhibit RhoA signaling, and modulate non-muscle myosin IIA serine phosphorylation. These results suggest that axon growth can be regulated by EphA3 in two ways: a ligand (ephrin)-dependent way that results in growth inhibition (as previously published), and a ligand-independent, EphA3 processing-dependent way that results in accelerated growth.

The presented data is generally of high technical quality and largely convincing. The topic is interesting, and the story quite well presented. However, several essential experiments are missing, and others could be improved. Moreover, in some cases, the conclusions are overstated.

Essential revisions:

1) The requirement of PS1 for axonal growth in vivo is not well documented. Determination of axon length based on neurofilament (NF) immunostaining of brain sections is not optimal. NF staining intensity appears to be reduced in immature neurons (Figure 1B). Hence, the reduction in NF staining in PS1^-/-^ tissue may let the axons appear shorter. A better approach would be length measurements of DiI-traced axons.

2) In Figure 2, the authors demonstrate the presence of an EphA3 CTF in PS1^-/-^ brains and take this as indirect evidence for lack of PS1 activity. The presence of the EphA3 ICD is not shown in vivo. Neither is the presence of the EphA3 ICD shown in cultured neurons. The only evidence for the generation of the EphA3 ICD is in EphA3-expressing HEK cells. This is a serious deficiency.

3) In Figure 3, overexpression of EphA3 ICD rescues axon growth defects in PS1^-/-^ neurons. The authors conclude that these results suggest that PS1-dependent "EphA3 ICD generation is required for axon growth" (subsection “EphA3 cleavage by PS1/γ-secretase mediates axon elongation”). This is wrong. These are gain-of-function experiments that demonstrate sufficiency, not necessity. I would argue that the authors should compare (in the same experiment) overexpression in PS1^-/-^ neurons of EphA3 ICD versus full-length EphA3, to make the point that the EphA3 ICD signals differently from full-length EphA3. Using PS1^-/-^ neurons is the better experiment than using DAPT. I would also request that the EphA3 shRNA knock down experiment includes the rescue with full-length EphA3 (panel E). It is expected that full-length EphA3 rescues the knock down of EphA3, but not, according to the authors' hypothesis, the lack of PS1.

4) In Figure 4, both EphA3 activation by ephrin-A5 and EphA3 knock down induce growth inhibition (panel B) to a similar extent. This is contradictory and should be discussed.

5) In Figure 5, EphA3 cleavage is proposed to mediate axon growth by inhibiting RhoA activity. The evidence is not strong. PS1 inhibition increases RhoA activity (panel E), reduces axon length and this effect can be rescued by Rho-kinase inhibitor (panel B). The authors fail to implicate EphA3 ICD in this process. They should demonstrate that the axon growth defect caused by PS1 inhibition is rescued by overexpression of EphA3 ICD, and that this brings RhoA activity down to normal levels. As control, the axon growth defect caused by PS1 inhibition is not rescued by overexpression of full-length EphA3, and this keeps RhoA activity high.

6) In Figure 6, EphA3 cleavage is proposed to mediate axon growth by regulating the phosphorylation of non-muscle myosin IIA. I find this part rather preliminary and not well connected to the rest of the story. Open questions are: How does EphA3 ICD increase serine phosphorylation of NMIIA? Is tyrosine kinase activity of EphA3 ICD required? How does RhoA activity tie in with NMIIA phosphorylation?

7) I find it unsatisfying that no attempts were made to provide in vivo evidence for this pathway. For example, could overexpression of EphA3 ICD by in utero electroporation of PS1^-/-^ embryos rescue the axon/neurite growth defects?

[Editors’ note: what now follows is the decision letter after the authors submitted for further consideration.]

Thank you for resubmitting your work entitled "Presenilin/γ-secretase-dependent EphA3 processing mediates axon elongation through non-muscle myosin IIA" for further consideration at *eLife*. Your revised article has been favorably evaluated by Huda Zoghbi (Senior Editor), a Reviewing Editor, and two reviewers.

The manuscript has been improved but there are some remaining issues that need to be addressed before acceptance, as outlined below:

*Reviewer #1:*

The authors satisfactorily addressed all my points. The revised manuscript has still however one point that needs some clarification:

In the in utero electroporation assays (Figure 3 D), the authors are supposed to measure the length of GFP-labelled axons (white). Can they ascertain that these are indeed axons and not leading processes of migrating cells?

*Reviewer #2:*

The authors have addressed all the queries and the paper should be accepted for publication.

*Reviewer #3:*

The authors have done a reasonable job in responding to my questions. The new results have confirmed their conclusions and I am almost satisfied.

There is, however, one remaining problem. The in utero electroporation assays were not done properly (although I agree they look promising). GFP-labeled axons were quantified at E16.5 in EphA3 ICD electroporated wild-type and *Psen1^-/-^* embryos. There is no evidence that, at this stage and in these experimental conditions, *Psen1* mutant axons are shorter than wild-type axons. This needs to be shown.

The authors should first establish that there is a phenotype at this stage of development, by quantifying control GFP electroporated axons from wild-type and *Psen1* mutants, then compare axon length between control GFP and EphA3 ICD-IRES-GFP electroporated *Psen1* mutants. The authors need to quantify a minimum of n=3 embryos per group.

---

## [Author Response]

[Editors’ note: the author responses to the first round of peer review follow.]

Reviewer #1:[…] However, in vivo evidence is lacking as most experiments were performed in vitro. Authors may at least discuss better and stress axon defect similarities, overlaps, and differences between presenilins and EphA3/ephrinAs mutant mice.

We thank the reviewer for these helpful comments. We have now added evidence for in vivo relevance of PS/secretase-dependent EphA3 cleavage in axon growth by performing DiI labeling and in utero electroporation assays in *Psen1*^+/+^ and *Psen1*^-/-^ embryos (Figure 3D; subsection “EphA3 cleavage by PS1/γ-secretase mediates axon elongation”; see response to point 7 of reviewer #3).

We have now described similarities and differences in axon morphology in *Psen1*^-/-^ and *Epha3/ephrinA* mutant mice in the Discussion section as follows:

“*Psen1*^-/-^ micedie after birth likely due to respiratory failures caused by incomplete lung expansion, which resembles the phenotype of *Epha3* null mice (Shen et al., 1997; Vaidya et al., 2003). Embryonic *Epha3* inactivation results in axon misrouting of callosal neurons, although no motor axon targeting abnormalities are observed in *Epha3^-^*^/-^ mice (Vaidya et al., 2003; Nishikimi et al., 2011). By contrast, *Epha3/Epha4* deletion causes disturbances in motor axon guidance and sensory-motor neuron assemblies (Gallarda et al., 2008).*”*

Reviewer #2:[…]1) The authors have only demonstrated the role of PS1 using PS1 knockout cells, although DAPT could also affect PS2 activity. Perhaps it would be interesting to use PS2 knockout cells or cells transfected with siRNA for PS2.

To examine the role of PS2 in axon growth, we analyzed axon length in wild-type (control) and *Psen2*^-/-^ neurons. As shown in Figure 1D, genetic *Psen2* knock-down does not affect axon length in cultured hippocampal neurons (subsection “Presenilin-1/γ-secretase is essential for axon elongation in vitro and in vivo”).

2) The study has been done in embryonic brains, are there also differences in axonal length/white matter density in older PS1 knockout mice?

This is a very intriguing and interesting question. *Psen1*^-/-^ mice die just after birth due to respiratory defects and *Psen1/2* double knockout mice show early embryonic lethality (Shen et al., 1997; Wong et al., 1997). The early postnatal lethality of *Psen1*^-/-^ mice precludes investigation of axon length/white matter density in older *Psen1*^-/-^ mice.

3) Although this study points to the interaction of the ICD with Rho signalling and NMIIA, it should also be ruled out that the effect is not mediated by transcriptional related mechanisms, as it occurs with other intracellular fragments resulting from PS1 cleavage, such as NICD and AICD.

We thank the reviewer for this relevant observation. It is well-established that PS1/γ-secretase-generated intracellular fragments, including those of APP, Notch1 or ErbB4, are localized at the nucleus where they act as transcriptional regulators (Carpenter, 2003; Lleó and Saura, 2011). It is possible that EphA3 ICD may affect transcriptional mechanisms during axon growth. This is a very interesting question that will obviously require further studies in the future. We sincerely believe that this is far from the scope of the present study but since this issue is very relevant we added a comment in the Discussion section as follows:

“It is also plausible that EphA3 ICD could mediate axon growth by acting through transcriptional mechanisms as described for APP and Notch ICDs (Lleó and Saura, 2011)”.

4) In Figure 5A-C the use of a Rho-kinase inhibitor does not seem to have any effect on axonal length in PS1 wild-type cells, by looking at the quantification shown in the graphs. Is this the case? What could be the interpretation? The sentence in subsection “EphA3 cleavage mediates axon elongation by inhibiting RhoA signalling” is a bit confusing and could be changed to "dominant-negative RhoA mutant (RhoA T19N) efficiently reversed axon length defects in PS1- and γ-secretase activity-deficient neurons".

This is a very interesting comment. Statistical analysis shows that there was an overall significant effect of Y27632 on axon growth in *Psen1*^+/+/-/-^ neurons (F(1,10)=33.82, *P* < 0.0002; Figure 5A) and Veh/DAPTtreated neurons (F(1,10)=11.64, *P* < 0.0066; Figure 5B). However, Bonferroni post-hoc analysis revealed no significant differences between vehicle and Y27632 treatments in control neurons (*P* > 0.05). This result including the interpretation is described in subsection “EphA3 cleavage mediates axon elongation partially by inhibiting RhoA signalling”.

Subsection “Ligand-independent EphA3 processing mediates axon elongation” has been modified.

5) The authors should propose a potential mechanism whereby the ICD affects NMIIA phosphorylation.

We have added a brief comment in the Discussion section suggesting that EphA3 ICD could affect NMIIA phosphorylation at Ser1943 by affecting activity and/or localization of the main NMIIA heavy chain kinases casein kinase II or PKC as follows:

“Future investigations are needed to uncover the mechanism by which EphA3 ICD enhances NMIIA phosphorylation. It is possible that EphA3 ICD could affect activity and/or localization of protein kinase C and casein kinase II, the main kinases that phosphorylates NMIIA heavy chain (Breckenridge et al., 2009). Interestingly, TGF-β increases NMIIA Ser1943 phosphorylation during epithelial-mesenchymal transition (Beach et al., 2011). Since TGF-β plays essential roles during neuron specification and activates RhoA-dependent signaling another possibility is that EphA3 ICD could regulate TGF-β signaling affecting RhoA and NMIIA phosphorylation”.

Reviewer #3:[…]Essential revisions:1) The requirement of PS1 for axonal growth in vivo is not well documented. Determination of axon length based on neurofilament (NF) immunostaining of brain sections is not optimal. NF staining intensity appears to be reduced in immature neurons (Figure 1B). Hence, the reduction in NF staining in PS1^-/-^ tissue may let the axons appear shorter. A better approach would be length measurements of DiI-traced axons.

We have performed retrograde DiI tracing in *Psen1^+/+^* and *Psen1^-/-^* brain sections. Quantitative analysis of DiI/neurofilament axons shows significant reduction of axons in *Psen1^-/-^* mouse brains. The new results are shown in Figure 1B and described in Results section (subsection “Presenilin-1/γ-secretase is essential for axon elongation in vitro and in vivo”) and the Figure 1 legend.

2) In Figure 2, the authors demonstrate the presence of an EphA3 CTF in PS1^-/-^ brains and take this as indirect evidence for lack of PS1 activity. The presence of the EphA3 ICD is not shown in vivo. Neither is the presence of the EphA3 ICD shown in cultured neurons. The only evidence for the generation of the EphA3 ICD is in EphA3-expressing HEK cells. This is a serious deficiency.

We thank the reviewer for this comment. Our results indicate that EphA3 is a putative PS/γ-secretase substrate as revealed by increased EphA3 CTF levels in *Psen1*^-/-^ brains and DAPT-treated *Psen1*^+/+^ neurons (Figure 2A,B). Biochemical detection of endogenous ICDs has been a difficult task for most of PS/γ-secretase substrates, even for the classical ones APP and Notch1 since these cleaved fragments are rapidly degraded o transported to the nucleus. For detecting intracellular fragments, γ-secretase in vitro assays were developed and extensively employed by different laboratories (Sastre et al., 2001). We used this approach to detect, purify and identify, by using mass spectrometry sequencing, the EphA3 ICD from EphA3expressing HEK2913 cells (Figure 2G,H; subsection “Presenilin-1/γ-secretase-dependent EphA3 cleavage”third paragraph).

We repeated the γ-secretase in vitro assay using embryonic (E17.5) mouse brains. Biochemical analysis shows the presence of EphA3 ICD (90 min) in the soluble fraction (S100) and EphA3 CTF in the pellet membrane fraction (P100) of brain lysates (see Author response image 1). This new result is shown in now shown in Figure 2G and Results section (subsection “Presenilin-1/γ-secretase-dependent EphA3 cleavage”).

**Author response image 1. respfig1:** Detection of EphA3 ICD in mouse brains in vivo. PS/γ-secretase-mediated EphA3 processing in mouse brain. In vitro γ-secretase activity assay in brain lysates of embryonic mouse brains (E17.5). Biochemical analysis shows generation of EphA3 ICD (90 min, arrowhead) in the soluble fraction (S100) and EphA3 CTF in the pellet fraction (P100). Mass spectrometry sequencing revealed the identity of this fragment as EphA3 ICD (Figure 2H).

3) In Figure 3, overexpression of EphA3 ICD rescues axon growth defects in PS1^-/-^ neurons. The authors conclude that these results suggest that PS1-dependent "EphA3 ICD generation is required for axon growth" (subsection “EphA3 cleavage by PS1/γ-secretase mediates axon elongation”). This is wrong. These are gain-of-function experiments that demonstrate sufficiency, not necessity. I would argue that the authors should compare (in the same experiment) overexpression in PS1^-/-^ neurons of EphA3 ICD versus full-length EphA3, to make the point that the EphA3 ICD signals differently from full-length EphA3. Using PS1^-/-^ neurons is the better experiment than using DAPT. I would also request that the EphA3 shRNA knock down experiment includes the rescue with full-length EphA3 (panel E). It is expected that full-length EphA3 rescues the knock down of EphA3, but not, according to the authors' hypothesis, the lack of PS1.

1) Sufficiency: As suggested by the reviewer, we modified this sentence as follows (subsection “EphA3 cleavage by PS1/γ-secretase mediates axon elongation”): “…suggesting that PS1/γ-secretase-dependent generation of EphA3 ICD is sufficient for axon growth”.

2) EphA3 ICD in *Psen1*^-/-^ neurons: We agree that experiments in *Psen1*^-/-^ neurons could be a better genetic assay than pharmacological inhibition. Since similar experiments were done in DAPT-treated neurons (Figure 3B) and due to the low number of available *Psen1*^-/-^ embryos in the lab (embryonic lethality, low availability of *Psen1*^+/-^ mice for crossings…), we prioritized the in vivo staining experiments suggested by the reviewer (DiI and in utero electroporation). Indeed, in utero electroporation assays revealed similar axon length in EphA3 ICD-transduced *Psen1*^+/+^ and *Psen1^-^*^/-^ brains after in utero electroporation assays (see below, Figure 3D).

3) Rescue with full-length EphA3: A rescue experiment with EphA3 fl and deletion mutants in *Epha3* shRNA neurons is shown in Figure 4D. Full-length EphA3, but not EphA3 deletion mutants, rescues axon defects in *Epha3* shRNA-transduced neurons (Figure 4D). Notably, EphA3 ICD, but not EphA3 fl or ΔICD, reversed axon length defects in PS/γ-secretase-deficient neurons (Figure 3B), which suggests that the effect of PS/γ-secretase on axon growth is mediated by EphA3 cleavage.

4) In Figure 4, both EphA3 activation by ephrin-A5 and EphA3 knock down induce growth inhibition (panel B) to a similar extent. This is contradictory and should be discussed.

Ephrin-A5 treatment inhibits axon growth by activating the classical ligand-EphA3 signaling pathway(Nishikimi et al., 2011), whereas *Epha3* silencing, which is expected to disrupt both the ligand dependent axon retraction and the ligand-independent PS/γ-secretase-mediated axon growth, causes an overall axon growth inhibition (Figure 4A,B,D). Interestingly, *Epha3* shRNA-induced axon defects are reversed by EphA3 ICD expression in hippocampal neurons (Figure 3C). These two alternative ligand-dependent and -independent pathways are illustrated in the summary scheme (Figure 7). We have now described these apparent contradictory results in the Discussion section as follows:

“…PS1/γ-secretase/EphA3-dependent axon growth contrasts with the classical role of ligand-induced EphA3 signaling in axon retraction. In agreement, both ligand-induced signaling and blocking ligand-independent EphA3 cleavage (e.g. *Epha3* silencing) inhibit axon growth suggesting that these two mechanisms occur in cellular conditions”.

5) In Figure 5, EphA3 cleavage is proposed to mediate axon growth by inhibiting RhoA activity. The evidence is not strong. PS1 inhibition increases RhoA activity (panel E), reduces axon length and this effect can be rescued by Rho-kinase inhibitor (panel B). The authors fail to implicate EphA3 ICD in this process. They should demonstrate that the axon growth defect caused by PS1 inhibition is rescued by overexpression of EphA3 ICD, and that this brings RhoA activity down to normal levels. As control, the axon growth defect caused by PS1 inhibition is not rescued by overexpression of full-length EphA3, and this keeps RhoA activity high.

To examine the role of PS1/γ-secretase and ICD on RhoA activity, we performed additional RhoA activity assays in primary neurons. These new results are described in subsection “EphA3 cleavage mediates axon elongation partially by inhibiting RhoA signalling”.

In agreement with a role of PS1/γ-secretase on axon growth through RhoA, DAPT increases RhoA activity in cultured neurons (Figure 5F) whereas pharmacological Rho inhibition (Y27632) reverses axon length defects in PS-deficient neurons (Figure 5A). By contrast, EphA3 ICD decreases RhoA activity in SK-N-AS neuronal cells (Figure 5E) and it rescues axon growth defects in *Psen1*^-/-^ and DAPT-treated neurons (Figure 3A, 3B, 5D). In primary neurons, however EphA3 ICD did not apparently affect RhoA activity in these experimental conditions (Figure 5F, right panel). This suggests that EphA3 ICD may affect RhoA activity differentially depending on the cell type or experimental conditions. We conclude that ¨ PS/γ-secretase-dependent EphA3 cleavage is unlikely to mediate axon elongation by affecting only RhoA signaling¨ (subsection “EphA3 cleavage mediates axon elongation partially by inhibiting RhoA signalling”).

6) In Figure 6, EphA3 cleavage is proposed to mediate axon growth by regulating the phosphorylation of non-muscle myosin IIA. I find this part rather preliminary and not well connected to the rest of the story. Open questions are: How does EphA3 ICD increase serine phosphorylation of NMIIA? Is tyrosine kinase activity of EphA3 ICD required? How does RhoA activity tie in with NMIIA phosphorylation?

See also reviewer #2 comment 5.

We agree that the molecular mechanism(s) linking EphA3 ICD with NMIIA heavy chain phosphorylation (Ser1943) is still an open question that should be addressed in future studies. Interestingly, a possible feedback loop between RhoA and NMIIA may exist in other cellular types (Hays et al., 2014), and NMIIA Ser1943 phosphorylation is up-regulated during TGF-β-induced epithelial-mesenchymal transition (Beach et al., 2011). We have now described this point in the Discussion section.

7) I find it unsatisfying that no attempts were made to provide in vivo evidence for this pathway. For example, could overexpression of EphA3 ICD by in utero electroporation of PS1^-/-^ embryos rescue the axon/neurite growth defects?

See also response to reviewer #1.

To study the in vivo relevance of PS/γ-secretase**-**dependent EphA3 signaling, we performed in utero electroporation assays by transducing EphA3 ICD-IRES-GFP vector in *Psen1*^+/+^ and *Psen1*^-/-^ mouse embryos. Our new results show no significant differences on axon length between EphA3 ICD-transduced *Psen1*^+/+^ and *Psen1*^-/-^ neurons (Figure 3D; *P* = 0.83; see subsection “EphA3 cleavage by PS1/γ-secretase mediates axon elongation”)

[Editors’ note: the author responses to the first round of peer review follow.]

The manuscript has been improved but there are some remaining issues that need to be addressed before acceptance, as outlined below:Reviewer #1:The authors satisfactorily addressed all my points. The revised manuscript has still however one point that needs some clarification:In the in utero electroporation assays (Figure 3 D), the authors are supposed to measure the length of GFP-labelled axons (white). Can they ascertain that these are indeed axons and not leading processes of migrating cells?

We thank the reviewer for this comment. As shown in Figure 1A-C using multiple axonal markers the neuronal processes affected by loss of *Psen1* in the ventricular zone, the same region analyzed in Figure 3D, are axons.

Reviewer #3:The authors have done a reasonable job in responding to my questions. The new results have confirmed their conclusions and I am almost satisfied.There is, however, one remaining problem. The in utero electroporation assays were not done properly (although I agree they look promising). GFP-labeled axons were quantified at E16.5 in EphA3 ICD electroporated wild-type and Psen1^-/-^ embryos. There is no evidence that, at this stage and in these experimental conditions, Psen1 mutant axons are shorter than wild-type axons. This needs to be shown.The authors should first establish that there is a phenotype at this stage of development, by quantifying control GFP electroporated axons from wild-type and Psen1 mutants, then compare axon length between control GFP and EphA3 ICD-IRES-GFP electroporated Psen1 mutants. The authors need to quantify a minimum of n=3 embryos per group.

We thank the reviewer for this important comment. As suggested by the reviewer we have performed in utero electroporation assays using control (mCherry) or EphA3ICD IRES-GFP vectors in multiple sections of *Psen1*^+/+^ and *Psen1*^-/-^ embryos. Embryonic brain sections (E16.5) were stained, imaged using laser confocal microscopy and stained processes were quantified. The new results (Figure 3D, subsection “EphA3 cleavage by PS1/γ-secretase mediates axon elongation”) reveal statistical significant differences (*P* < 0.0001) in axon length in *Psen1*^-/-^ embryos compared to *Psen1*^+/+^ at E16.5 stage. Moreover, EphA3 ICD efficiently reversed axon length defects in *Psen1*^-/-^ brains (EphA3 ICD: *Psen1*^-/-^ vs. *Psen1*^+/+^, *P* = 0.61).